# Spatiotemporal tissue maturation of thalamocortical pathways in the human fetal brain

Siân Wilson[1,2], Maximilian Pietsch[1], Lucilio Cordero-Grande[1,3,4], Daan Christiaens[1,5], Alena Uus[6], Vyacheslav R Karolis[1], Vanessa Kyriakopoulou[1], Kathleen Colford[1], Anthony N Price[1], Jana Hutter[1], Mary A Rutherford[1], Emer J Hughes[1], Serena J Counsell[1], Jacques-Donald Tournier[1], Joseph V Hajnal[1], A David Edwards[1,2], Jonathan O'Muircheartaigh[1,2,7,8]*, Tomoki Arichi[1,2,9,10]*

[1]Centre for the Developing Brain, School of Biomedical Engineering and Imaging Sciences, King's College London, London, United Kingdom; [2]Centre for Neurodevelopmental Disorders, King's College London, London, United Kingdom; [3]Biomedical Image Technologies, ETSI Telecomunicación, Universidad Politécnica de Madrid, Madrid, Spain; [4]Biomedical Research Networking Center in Bioengineering, Biomaterials and Nanomedicine (CIBER-BBN), Madrid, Spain; [5]Department of Electrical Engineering (ESAT/PSI), Katholieke Universiteit Leuven, Leuven, Belgium; [6]Department of Biomedical Engineering, School Biomedical Engineering and Imaging Sciences, King's College London, St. Thomas' Hospital, London, United Kingdom; [7]Department of Forensic and Neurodevelopmental Sciences, King's College London, London, United Kingdom; [8]Department of Neuroimaging, Institute of Psychiatry, Psychology and Neuroscience, King's College London, London, United Kingdom; [9]Children's Neurosciences, Evelina London Children's Hospital, Guy's and St Thomas' NHS Foundation Trust, London, United Kingdom; [10]Department of Bioengineering, Imperial College London, London, United Kingdom

*For correspondence:
jonathanom@kcl.ac.uk (JO'M);
tomoki.arichi@kcl.ac.uk (TA)

Competing interest: The authors declare that no competing interests exist.

**Abstract** The development of connectivity between the thalamus and maturing cortex is a fundamental process in the second half of human gestation, establishing the neural circuits that are the basis for several important brain functions. In this study, we acquired high-resolution in utero diffusion magnetic resonance imaging (MRI) from 140 fetuses as part of the Developing Human Connectome Project, to examine the emergence of thalamocortical white matter over the second to third trimester. We delineate developing thalamocortical pathways and parcellate the fetal thalamus according to its cortical connectivity using diffusion tractography. We then quantify microstructural tissue components along the tracts in fetal compartments that are critical substrates for white matter maturation, such as the subplate and intermediate zone. We identify patterns of change in the diffusion metrics that reflect critical neurobiological transitions occurring in the second to third trimester, such as the disassembly of radial glial scaffolding and the lamination of the cortical plate. These maturational trajectories of MR signal in transient fetal compartments provide a normative reference to complement histological knowledge, facilitating future studies to establish how developmental disruptions in these regions contribute to pathophysiology.

## Editor's evaluation

This study presents important new findings regarding prenatal thalamocortical development. The authors present convincing evidence, while overcoming substantial methodological challenges, in

charting prenatal brain development in vivo. This work will be of interest to pediatric and developmental neuroscientists and neuroradiologists.

## Introduction

Thalamocortical connections provide important inputs into the developing cortex during the second half of human gestation, where they play a key role in guiding cortical areal differentiation and establishing the circuitry responsible for sensory integration across the lifespan (*Jones, 2007*; *Price et al., 2006*; *Schummers et al., 2005*; *Sharma et al., 2000*; *Sur and Rubenstein, 2005*). Their importance is highlighted by previous work implicating disruptions to thalamocortical development during the perinatal period in the pathophysiology of neurodevelopmental disorders, such as schizophrenia (*Klingner et al., 2014*; *Marenco et al., 2012*), bipolar disorder (*Anticevic et al., 2014*), and autism (*Nair et al., 2013*). Altered thalamocortical connectivity has also been described in preterm infants, and was used to predict cognitive outcome (*Ball et al., 2013*; *Ball et al., 2015*; *Toulmin et al., 2021*), highlighting the specific vulnerability of these pathways during the second to third trimester. Although thalamocortical development has been studied in rodents and non-human primates (*Brody et al., 1987*; *Kostović and Jovanov-Milosević, 2006*; *Molnár and Blakemore, 1995*; *Yakovlev et al., 1960*) and post-mortem human tissue (*Krsnik et al., 2017*; *Takahashi et al., 2012*; *Wilkinson et al., 2017*), little is known about in vivo white matter maturation during fetal development.

White matter development in the late second and third trimesters of human gestation (between 21 and 37 weeks) is characterised by a sequence of precisely timed biological processes occurring in transient compartments of the fetal brain. These processes include the migration of neurons along the radial glial scaffold, accumulation of thalamocortical axons in the superficial subplate, innervation of the target cortical area, conversion of radial glial cells into astrocytes, and ensheathment of axonal fibres (*Krsnik et al., 2017*; *Molliver et al., 1973*; *Kostović et al., 2002*, *Kostovic and Judas, 2006*). The challenge for in vivo neuroimaging studies is to disentangle the effect of these different neurobiological processes on the diffusion magnetic resonance imaging (dMRI) signal, to improve mechanistic insight about the transformation of transient fetal compartments into segments of developing white matter (Kostovic 2012).

Recent advances in diffusion weighted imaging now allow in vivo characterisation and estimation of white matter development during the fetal period. Tractography has been used to estimate the fetal brain's major white matter bundles and quantitatively characterise the evolution of the microstructure across the second half of gestation (*Bui et al., 2006*; *Zanin et al., 2011*; *Jaimes et al., 2020*; *Jakab et al., 2015*; *Keunen et al., 2018*; *Khan et al., 2019*; *Machado-Rivas et al., 2021*; *Wilson et al., 2021*). Advanced acquisition and analysis methods enable the relative contribution of constituent tissue and fluid compartments to the diffusion signal to be estimated (*Jeurissen et al., 2014*; *Pietsch et al., 2019*). Using this approach, previous work has identified non-linear trends in diffusion metrics over the second to third trimester (*Wilson et al., 2021*). Namely, we observed an initial decrease in tissue fraction within developing white matter between 22 and 29 weeks, which could be due to the radial glial scaffold disassembling (*Rakic, 2003*). Subsequently, we observed an increase from 30 to 36 weeks, potentially linked to more coherent fibre organisation, axonal outgrowth, and ensheathment (*Wimberger et al., 1995*; *Back et al., 2002*; *Haynes et al., 2005*), increasing the structural integrity of maturing white matter. Interpreting these trends is especially challenging in the rapidly developing fetal brain, because of the high sensitivity and low specificity of diffusion metrics to various co-occurring biological processes.

We hypothesise that the biological processes occurring in different fetal compartments leads to predictable changes in diffusion metrics along tracts, reflecting the appearance and resolution of these transient zones. When a mean value across the whole tract is calculated, sensitivity to the unique neurobiological properties of each transient compartment is lost. For example, in the early prenatal and mid prenatal period, the subplate is a highly water-rich compartment containing extracellular matrix, whereas the cortical plate and the deep grey matter are relatively cell dense (*Kostović, 2020*). We therefore predict that the tissue fraction would be higher in the deep grey matter and the cortical plate and lower in the subplate. We investigate this by characterising the entire trajectory of tissue composition changes between the thalamus and the cortex, to explore the role of transient fetal brain developmental structures on white matter maturational trajectories.

We acquired diffusion weighted imaging from 140 fetuses over a wide gestational age (GA) range (21–37 weeks) and use tractography to delineate five distinct thalamocortical pathways. To investigate whether the immature axonal bundles can be traced back to specific and distinct locations within thalamus, we parcellate the thalamus according to streamline connectivity (*Behrens et al., 2003*). We find consistent and distinct origins of different tracts, resembling the adult topology of thalamic nuclei (*Toulmin et al., 2015*; *Behrens et al., 2003*) as early as 23 weeks' gestation. We then apply a multi-shell multi-tissue constrained spherical deconvolution (MSMT-CSD) diffusion model (*Jeurissen et al., 2014*) and derive tissue and fluid fraction values, charting tract-specific maturational profiles over the second to third trimester. We overlay the tracts on an atlas of transitioning fetal compartments and correlate changes in the dMRI signal across time with critical neurodevelopmental processes, such as the dissolution of the subplate and lamination of the cortical plate. We demonstrate that along-tract sampling of diffusion metrics can capture temporal and compartmental differences in the second to third trimester, reflecting the maturing neurobiology of the fetal brain described in histological studies. With these methods, we provide a detailed, accurate reference of the unique developing microstructure in each tract that improves mechanistic insight about fibre maturation, bridging the gap between MRI and histology.

## Results

### Estimating thalamocortical pathways using probabilistic streamline tractography

High-angular-resolution multi-shell diffusion weighted imaging (HARDI) was acquired from 140 fetuses between 21 and 37 gestational weeks (70 male, 70 female) as part of the Developing Human Connectome Project (dHCP). Data were corrected for fetal head motion and other imaging artefacts (*Christiaens et al., 2021*). Individual subject orientation density functions (ODFs) were then computed using cohort-specific fluid and 'tissue' response functions and compiled to generate weekly diffusion templates (see Materials and methods). The diffusion templates were then registered to a T2-weighted brain atlas (*Gholipour et al., 2017*) of tissue segmentations, used to generate anatomically constrained whole-brain connectomes for each gestational week

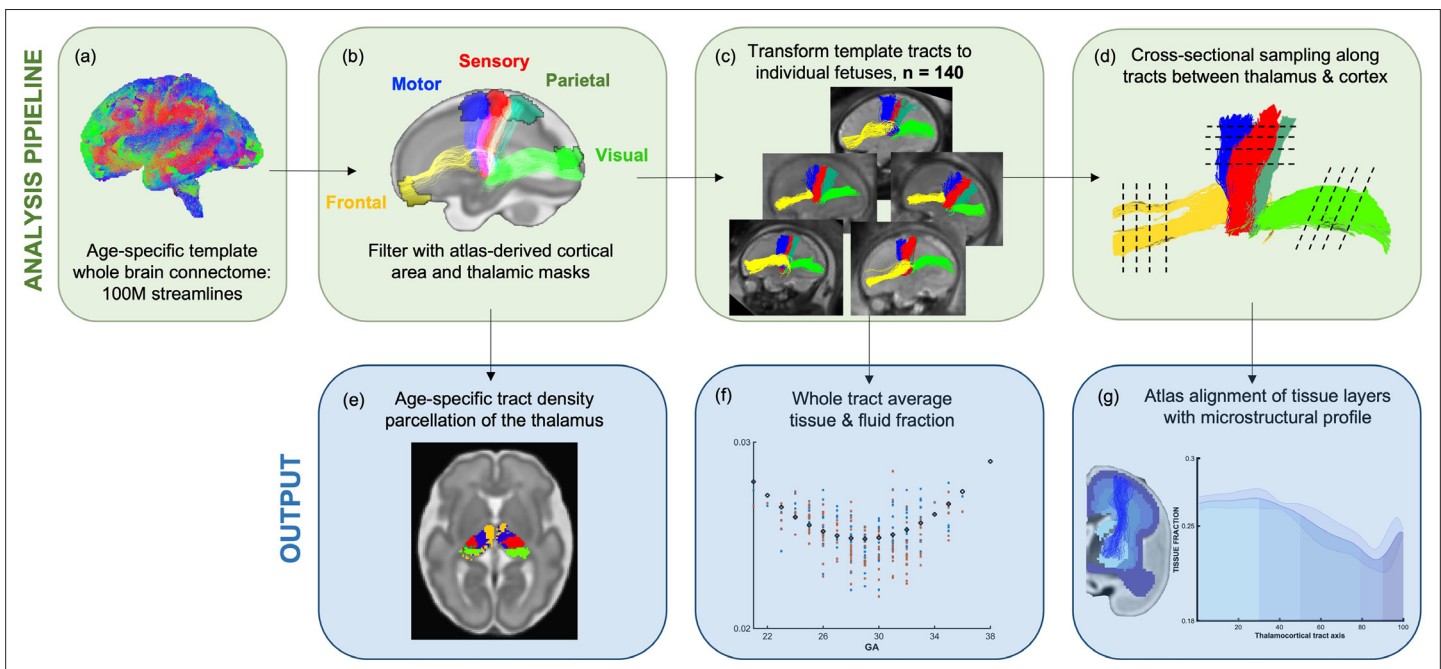

**Figure 1.** Methods pipeline to estimate and quantify thalamocortical tracts development. (Top row) (**a**) Whole-brain connectomes generated for each gestational week template. (**b**) Atlas-defined masks of the thalamus and cortical areas were used to extract white matter pathways of interest from the connectomes. (**c**) These pathways were transformed to the native fetal diffusion space, (**d**) the values were sampled along the tract. (**f**) Whole-tract average diffusion metrics were calculated or (**g**) values sampled along the tract were aligned to an atlas of transient fetal compartments.

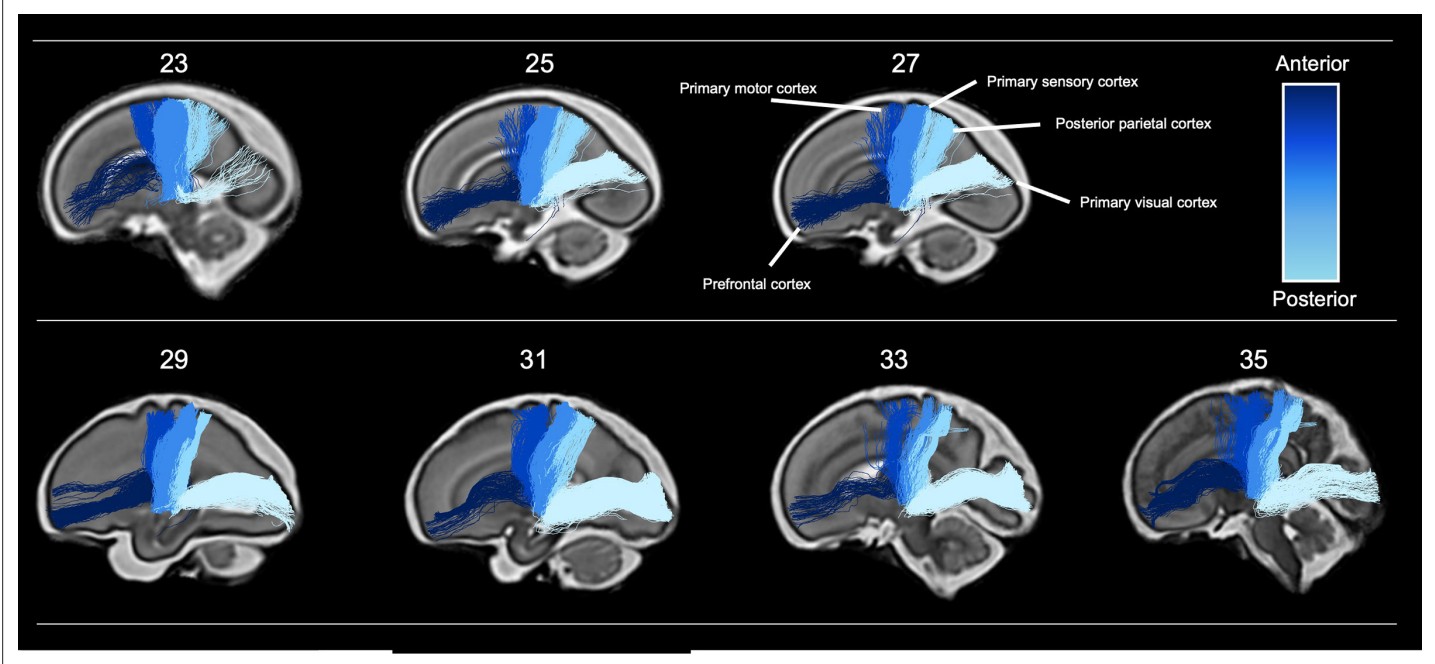

**Figure 2.** Tractography of thalamocortical pathways in different gestational week templates across the second to third trimester. Tracts project to five different cortical areas, the prefrontal cortex, primary motor cortex, primary sensory cortex, posterior parietal cortex, and primary visual cortex, coloured according to the anterior-posterior axis.

(*Smith et al., 2012*; *Tournier et al., 2019*). To constrain our investigation, we selected thalamo-cortical pathways that are at a critical stage in their development and are vulnerable to external influences in the second to third trimester (*Batalle et al., 2017*; *Nosarti et al., 2014*; *Raybaud et al., 2013*), the anterior thalamic radiation (AT), thalamic-motor tract (TM), thalamic-sensory tract (TS), posterior parietal tract (PP), and optic radiation (OR). The connectomes were filtered down to the pathways of interest using inclusion regions defined by the T2 atlas, including the thalamus and specific cortical areas (*Figure 1*). These included the primary motor cortex, primary sensory cortex, posterior parietal cortex, dorso-lateral prefrontal cortex, and the primary visual cortex. With this method, we were able to delineate five major thalamocortical pathways in each gestational week. To keep regions of interest more consistent across the cohort, we grouped all cases into 2-weekly intervals, starting at 23 weeks (*Figure 2*), replicating methods used previously (*Wilson et al., 2021*).

## Structural connectivity parcellation of the fetal thalamus resembles adult topology of thalamic nuclei

Tract density imaging (*Calamante et al., 2010*) was used in each ODF template to explore whether the different cortical areas were connected to distinct, specific regions of the thalamus (*Figure 3a*). We found that for all ages, there was symmetrical topographical representation of the cortical regions of interest in the thalamus. Furthermore, they spatially corresponded to the adult organisation of thalamic nuclei, demonstrated by the schematic (*Figure 3a*) which is based on Morel's thalamus and other connectivity-derived parcellations from adult imaging studies (*Morel et al., 1997*; *Najdenovska et al., 2018*; *Niemann et al., 2000*). The tract projecting to the prefrontal cortex was connected to the anterior thalamus and in the younger ages (23–29 weeks) also to the medial thalamus. In the older templates (31, 33, and 35 weeks), frontal connectivity was more localised to the anterior thalamus and less evident in the medial area. There were distinct but neighbouring areas in the ventral thalamus connecting to the sensory and motor cortical areas, the motor-connected thalamic region being more frontal. The connectivity of the posterior parietal area was in the posterior part of the thalamus, and the most posterior voxels in the thalamic mask projected to the primary visual cortex.

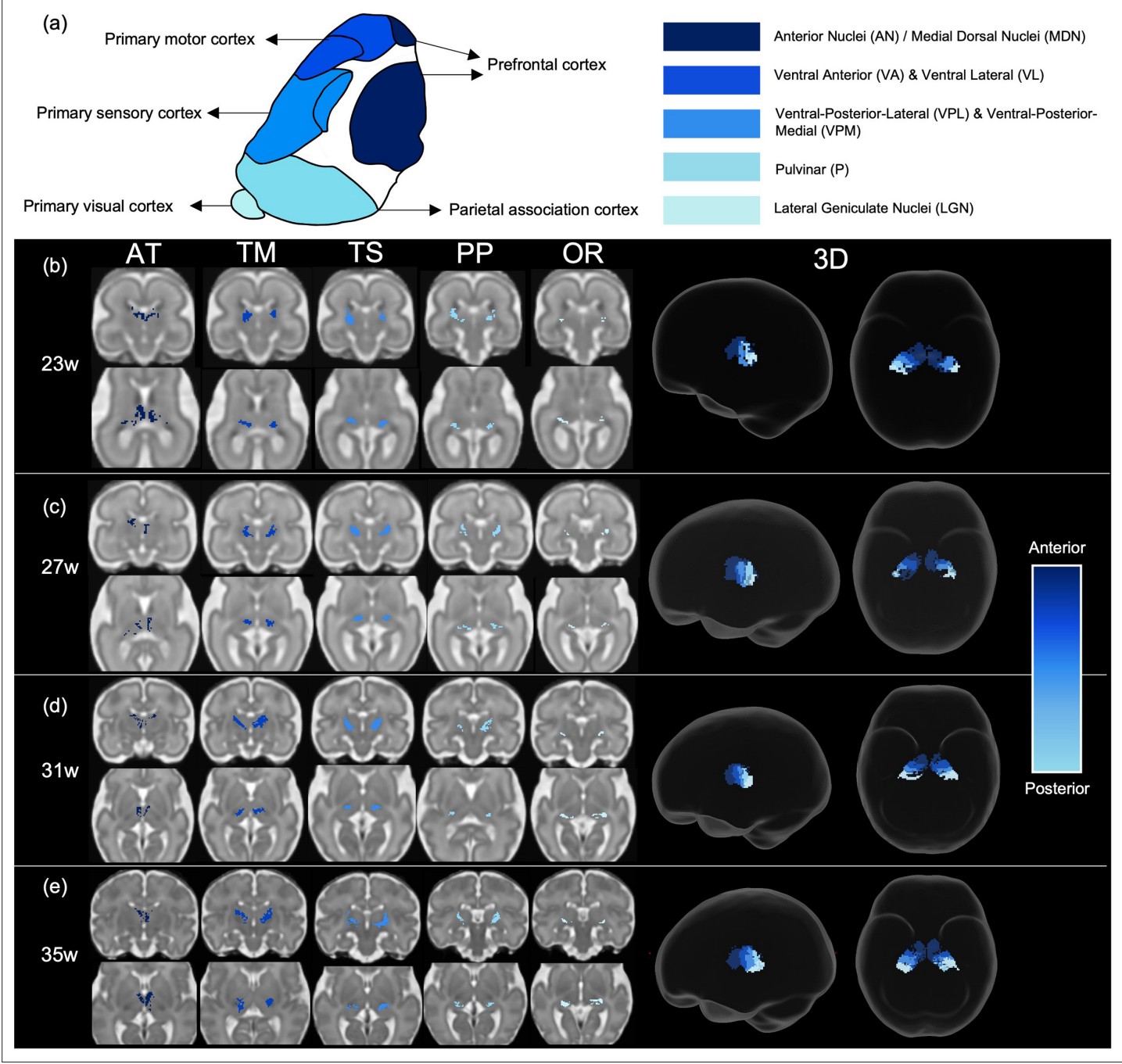

**Figure 3.** Tract density imaging parcellation of thalamus at different fetal ages. (**a**) A schematic of expected cortical connectivity arrangement across the thalamus, based on Morel's parcellation of the adult thalamic nuclei. (**b**) Axial slices of thalamic parcellation, thresholded for the top 20% of voxels, colour-coded according to streamline connectivity of different tracts at 23 weeks, (**c**) 27 weeks, (**d**) 31 weeks, and (**e**) 35 weeks.

## Whole-tract average diffusion metrics have a characteristic U-shaped trend across the second to third trimester

The thalamocortical pathways were transformed from the age-matched templates to the native subject space for 140 fetal subjects (*Figure 4a*). The MSMT-CSD-derived voxel-average tissue and fluid ODF values were sampled along the warped group-average streamline tracts. Tract-specific values were derived by averaging these for each tract in each subject, replicating the approach that has been used in previous fetal studies (*Wilson et al., 2021*). The values for each tract were plotted against the GA

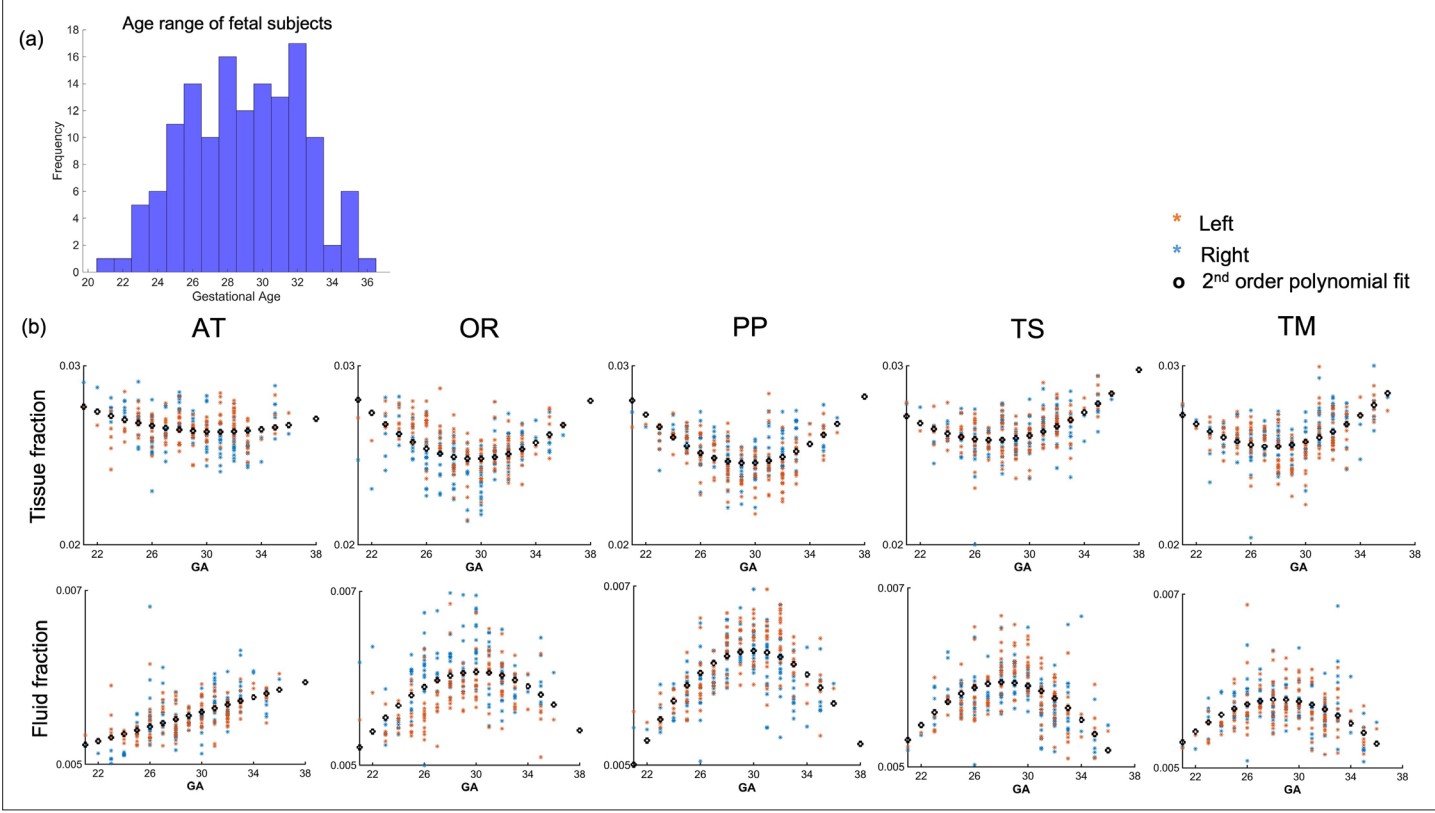

**Figure 4.** Diffusion metric age trajectories for each tract. (**a**) Distribution of age among the fetal cohort (n=140) in gestational weeks. (**b**) Whole-tract average tissue (top) and fluid fractions (bottom) for each subject in the left (orange) and right (blue) hemisphere, plotted against gestational age (GA) of the subject, best fit by second-order polynomials (AT = anterior thalamic radiation, OR = optic radiation, PP = posterior parietal tract, TS = thalamic-sensory tract, TM = thalamic-motor tract).

The online version of this article includes the following figure supplement(s) for figure 4:

**Figure supplement 1.** Whole-tract average of diffusion tensor metrics in the fetal cohort (n=140), fractional anisotropy (FA), and mean diffusivity (MD), in thalamocortical tracts across gestational age (every other week shown).

---

of the subject. The Akaike information criterion suggested second-order polynomial relationships for all tracts for both tissue and fluid fraction metrics, except the fluid fraction in the AT which is linear (*Figure 4b*). Diffusion tensor metrics also displayed similar age-related polynomial trends (*Figure 4—figure supplement 1*). We compared male vs. female (two-sample test) in each gestational week and found no significant differences (p>0.1).

## Along-tract sampling reveals evolving properties of fetal brain transient compartments

To explore the origins of these trends in diffusion metrics, the values of tissue and fluid fraction were sampled in subject space at 100 equidistant intervals between the thalamus and the cortex. Tissue and fluid fraction are scaled jointly per scan such that they are approximately reciprocal of one another across the brain using a cubic polynomial spatial model (*Pietsch et al., 2019*). In each subject, we sampled the tissue and fluid fraction values beneath the streamlines from the thalamus to the cortex, plotting the microstructural tissue composition against the distance from the thalamus (*Figure 5*). We found that trajectories changed gradually between gestational weeks, and therefore we grouped them to match previous histological studies that define this fetal period according to three developmental windows, early (21–25.5 weeks), mid (26–31.5 weeks), and late (32–36 weeks) prenatal period (*Kostović, 2020*; *Appendix 1—figures 5 and 6*). When comparing the microstructural profiles of all the tracts in the different periods, the motor, sensory, and parietal tracts shared similar trajectories, whilst those in the AT and OR tracts were more distinct (*Figure 5a, b, and c* and *Figure 5—figure supplement 1a,b*).

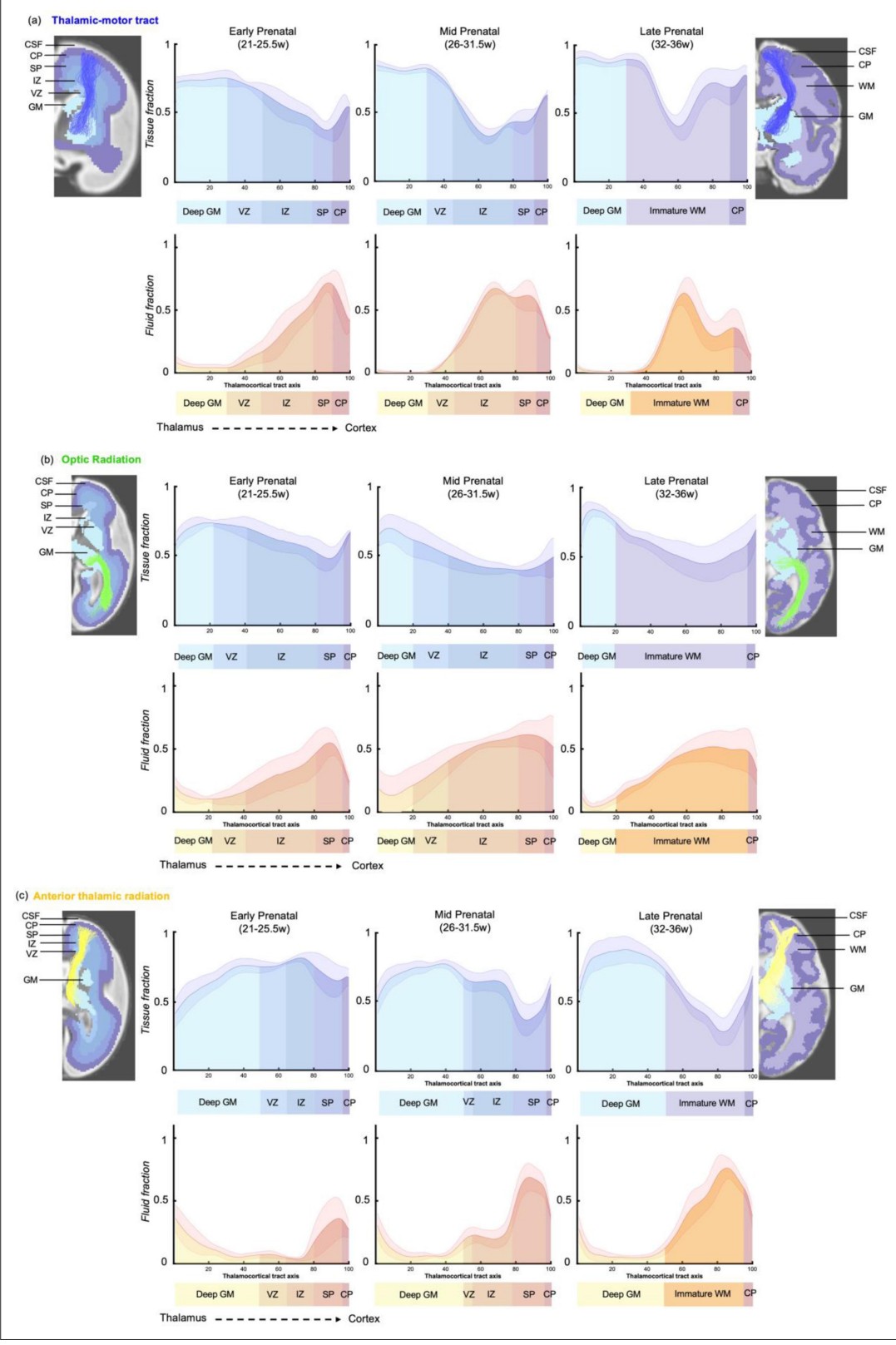

**Figure 5.** Microstructural composition of fetal compartments traversed by developing thalamic white matter. Tracts were overlayed on the atlas of fetal compartments (examples highlight the difference between fetal brain structure in early prenatal [25 weeks] on far left, and late prenatal [35 weeks] on far right). Tissue fraction trends (top row) and fluid fraction trends (bottom row), normalised to 1, between the thalamus and cortex (thalamocortical tract axis)

*Figure 5 continued on next page*

*Figure 5 continued*

for the (**a**) thalamic-motor tract, (**b**) optic radiation, and (**c**) anterior thalamic radiation. Subjects were grouped by age, and average trajectories plotted for early prenatal (22–25.5 weeks), mid prenatal (26–31.5 weeks), late prenatal (32–36 weeks). Error bars represent the standard deviation among all subjects in each group. Atlas-derived tissue boundaries are marked on the trajectories to reveal the changing tissue properties of each layer between early, mid, and late prenatal development (cortical spinal fluid = CSF, cortical plate = CP, subplate = SP, intermediate zone = IZ, ventricular zone = VZ, deep grey matter = GM, immature white matter = WM).

The online version of this article includes the following figure supplement(s) for figure 5:

**Figure supplement 1.** Microstructural composition of fetal compartments traversed by developing thalamic white matter.

To improve our ability to corroborate changes in the dMRI signal with observations from histological studies, we mapped the maturational trajectories to an atlas of fetal brain compartments (*Gholipour et al., 2017*) and overlayed the boundaries of these compartments on the tissue and fluid fraction trajectories (*Figure 5*). By overlaying the tracts on the atlas, and corroborating each sampling segment with an atlas region, we could characterise the diffusion properties of the different fetal compartments independent of their change in size across gestation. Tissue fraction values in the deep grey matter and the cortical plate areas increased with GA in all tracts. This increase was most marked in the tracts terminating in superior areas of the brain (motor, sensory, and superior parietal cortex) (*Figure 5a* TM, *Figure 5—figure supplement 1* (a) TS and (b) PP). The tissue fraction of the ventricular and intermediate zones decreased between the early and mid prenatal period, in all tracts. This decrease was very pronounced in the motor, sensory, and superior parietal tracts. The subplate tissue fraction changes were more tract specific. In the subplate of sensorimotor and parietal tracts, there was initially a very high fluid fraction and low tissue fraction, which transitions across the second to third trimester, increasing in tissue fraction from early to mid and then to late prenatal. Whereas in the AT, there was a decrease in subplate tissue fraction with GA (and a reciprocal increase in fluid fraction). In the OR, the subplate tissue fraction decreases between early and mid prenatal to then increase again in late prenatal. Highest tissue fractions were generally observed in the ventricular zone, with the lowest tissue fraction in the subplate area. To statistically test if there was a difference between the values in each compartment across GA, we correlated the tissue fraction values against age for each of the 100 sampling points. After correcting for multiple comparisons, points 1–16, 24–42, and 70–100 had significant linear correlations with age. Sampling points 43–69 had significant second-order polynomial relationships with age.

## Discussion

In this work, we studied in utero development of five distinct thalamocortical pathways using state-of-the-art dMRI methods and bespoke pre-processing pipeline (*Christiaens et al., 2019a*; *Cordero-Grande et al., 2019*; *Hutter et al., 2018a*; *Pietsch et al., 2019*; *Wilson et al., 2021*) in 140 fetuses aged 21–37 weeks' gestation. We show that these pathways connect to distinct thalamic nuclei, which could be clearly defined at group level even at 23 weeks. To disentangle the impact of different neurobiological processes on diffusion metrics, we characterised the tissue composition profile along each of the thalamocortical tracts as they traverse the different developmental tissue layers of the fetal brain. We used MSMT-CSD to model this fetal DWI dataset because it does not mandate a specific set of b-values. The technique exploits the unique b-value dependencies of different tissue types, and so depends inherently on the characteristics of the tissue in the fetal brain. The distinct properties of different fetal compartments after the application of MSMT-CSD are highlighted by the ODFs (*Appendix 1—figure 4*). Readers are directed to *Tournier et al., 2019*, for a more comprehensive review of the optimisation process involved to select appropriate b-shell values for the acquisition.

We found that the spatiotemporal changes in the diffusion signal reflected known developmental processes that take place between the early, mid, and late prenatal period. The early period is characterised by higher tissue fractions in the middle of the tract, where there is a radial scaffold for migrating neurons. As this scaffold dissipates in the mid prenatal period, this is accompanied by a reduction in the tissue fraction in the middle of the tract, and an increase in tissue fraction towards the termination of the tracts as the neurons of the cortical plate mature. Finally in the late prenatal period,

we observe the highest tissue fraction values at the start and end of the axis, as the pre-myelination phase of white matter development commences. This study demonstrates how the dMRI signal can be modelled to create in vivo spatiotemporal trajectories which relate to underlying neurobiological properties and are consistent with described trends from post-mortem histology (*Kostović, 2020*).

Early embryonic patterning of gene expression and cell division in the thalamus provide a template for specialised nuclei to emerge over the course of development, such that specific cells eventually occupy distinct locations within the thalamus (*Clascá et al., 2012*; *Nakagawa, 2019*, *Kostovic and Goldman-Rakic, 1983*). Rodent studies labelling the embryonic thalamus demonstrated that there is a characteristic topography of thalamic projections which roughly exists at their time of arrival; anterior to posterior movement along the convexity of the cortex is represented in a medial-to-lateral axis within the thalamus, while ventral-dorsal movement across the cortex is represented in an anterior-to-posterior axis within the thalamus (*Molnár et al., 1998*; *Molnár et al., 2012*). Thalamocortical tracts emerge over the same timescale as the thalamus parcellates and matures into its specialised group of nuclei (*Clascá et al., 2012*). Although the topography of thalamic nuclei and their cortical connectivity is acquired embryonically, no in vivo parcellation of the thalamus in the fetal brain has been published. Using tract density imaging, we observed that the cortical areas were connected to specific thalamic regions, organised in an anterior-posterior axis. This anterior-posterior representation of cortical connectivity in the thalamus was consistent across the second to third trimester and is in accordance with the topology of thalamic nuclei described in rodent studies and histology (*Molnár and Blakemore, 1995*; *Molnár et al., 1998*).

In addition, our fetal structural connectivity parcellation resembles the functionally derived thalamic parcellation in neonates, supporting the view that there is a strong association between structure and function in thalamocortical circuitry that begins early in life (*Johansen-Berg et al., 2005*; *Toulmin et al., 2015*; *Alcauter et al., 2014*). However, it is worth noting that this thalamic parcellation is dependent on streamline count through a voxel, and in the fetal brain streamlines are prone to spurious detection. This limitation is particularly relevant in the youngest fetuses, where we observe an extremely dense connectome (due to a fixed number of streamlines in a smaller brain) but there are very few coherent axonal bundles, so tracts might be overrepresented in the thalamic parcellation. The topography of thalamic nuclei is also not static during the embryonic and fetal period (*Le Gros Clark, 1936*), as pulvinar size increases and the dorsal lateral geniculate nucleus shifts its position from dorsolateral to ventromedial (*Rakic, 1977*; *Mojsilović and Zecević, 1991*). However, the image resolution of this study and the timespan in development over which this data was collected limit us from visualising these differences across age.

Recent studies characterising developing white matter pathways using human fetal MRI identified second-order polynomial maturational trends in diffusion metrics unique to this developmental period (*Wilson et al., 2021*; *Machado-Rivas et al., 2021*). Here, we replicated these methods with a different group of tracts and found the same U-shaped trends in thalamocortical white matter development. The inflection point at around 29–30 weeks was hypothesised to be the result of the dissipating radial glial scaffold followed by the pre-myelination phase of white matter development (*Wilson et al., 2021*; *Machado-Rivas et al., 2021*). The sensitivity of HARDI to radially organised structure in the fetal brain has been described by previous studies (*Miyazaki et al., 2016*; *Takahashi et al., 2012*; *Xu et al., 2014*) combining it with post-mortem tissue analysis to show that radially coherent diffusion signal corresponded to radial glial fibres in the early prenatal period, transitioning to cortico-cortical fibres around 30 weeks, coinciding with the appearance of astrocytes (*Takahashi et al., 2012*; *Xu et al., 2014*). However, with whole-tract average values, it is not possible to establish the precise effect of different neurodevelopmental processes on diffusion metrics across gestation.

To address this ambiguity, we characterised the entire trajectory of tissue composition changes between the thalamus and the cortex. We found that age-related changes in the tissue and fluid fraction along the tracts concurred with histological observations (*Kostović and Judas, 2010*). During the early prenatal period (22–25.5 GW), neuronal precursors migrate along the radial glial scaffold from proliferative zones to their destination in the cortical plate and thalamocortical axons accumulate in the superficial subplate, entering a 'waiting phase', forming transient synaptic connections (*Ghosh et al., 1990*; *Kostovic and Rakic, 1984*; *Kostovic and Rakic, 1990*). In terms of the diffusion signal, this strongly aligned microstructure of the radial glia is represented in our results by a higher tissue fraction in the transient compartments containing the most migratory cells (such as the VZ, IZ)

(*Kostović and Judas, 2010*). Conversely, we observe the lowest tissue fraction in the early prenatal SP, which predominantly contains hydrophilic extracellular matrix, as demonstrated by rodent and non-human primate studies (*Allendoerfer and Shatz, 1994*; *Miller et al., 2014*; *Molnár and Hoerder-Suabedissen, 2016*).

By the mid prenatal period (26–31.5 weeks), we observe increased tissue fraction in the cortical plate, coinciding with the innervation of the cortical plate by thalamocortical axons, increasing soma volume and dendritic branching of CP neurons and CP synaptogenesis (*Huttenlocher, 1979*; *Mrzljak et al., 1992*; *Huttenlocher and Dabholkar, 1997*). We also observe increased tissue fraction in the SP zone in the mid prenatal period, consistent with histological observations of increased coherence of axonal fibres between cortical areas (*Takahashi et al., 2012*; *Xu et al., 2014*). The tissue fraction in the VZ and IZ decreases compared to the early prenatal period, corresponding to the timeframe when the radial glial scaffold dissipates (*Kinney et al., 1988*; *Back et al., 2001*; *Haynes et al., 2005*).

From the mid to late prenatal period, there is a marked increase in tissue fraction in last third of the axis between thalamus and cortex. By this point in development, the radial glia have converted into oligodendrocyte precursor cells which ensheath the axonal fibres to commence pre-myelination, enhancing the structural integrity of the fibre pathways (*Back et al., 2001*; *Back et al., 2002*; *Haynes et al., 2005*; *Kinney et al., 1988*; *Kinney et al., 1994*). A previous study in perinatal rabbits has shown that this oligodendrocyte lineage progression correlates with diffusion metrics (*Drobyshevsky et al., 2005*), suggesting it is likely to contribute to the increased tissue fraction we observe in the late prenatal period. The tissue fraction increase in the CP area is consistent in time with the lamination of the CP, the elaboration of thalamocortical terminals in layer IV, and a rapid growth of basal dendrites of layer III and V pyramidal neurons (*Kostović and Jovanov-Milosević, 2006*; *Krsnik et al., 2017*; *Molliver et al., 1973*). These high tissue fraction values at the origin and termination of the tracts suggest co-maturation between ascending and descending pathways between the thalamus and cortex to eventually form continuous, structurally mature fibre bundles. This concept was proposed in the 1990s by Blakemore and Molnar, termed the 'handshake hypothesis'. They suggested that thalamocortical pathways ascending through the internal capsule project to their cortical targets with assistance from reciprocal descending cortical pathways (*Molnár and Blakemore, 1995*). We hypothesise that continuing this analysis over subsequent weeks into the neonatal period would lead to an increasing tissue fraction in the middle of the axis, as fibre bundles become more uniformly structurally mature and the subplate completely resolves (*Kinney et al., 1988*; *Haynes et al., 2005*; *Kostović and Jovanov-Milosević, 2006*).

We observed that tracts terminating superiorly (motor, sensory, and parietal) shared very similar trajectories in the early, mid, and late periods. However, the OR and the AT had more distinct trajectories. The microstructural change along the AT suggests increasing tissue fraction between the deep grey matter, VZ and IZ. We hypothesise that the high tissue fraction in the IZ is due to densely packed ascending and descending bundles within the anterior limb of the internal capsule (*Emos et al., 2022*). The precise timeline for outgrowth and intermingling of the thalamocortical and corticothalamic projections remains ambiguous, and appears to be different between rodents and non-human primates (*Alzu'bi et al., 2019*).

On the other hand, the OR traverses the deep parietal lobe along the border of the lateral ventricle and has smoother transitions in tissue fraction between the fetal compartments. This is likely due to the tract area running more parallel to the tissue interfaces. Another explanation for the regional differences in microstructural properties is the variation in subplate remnants. In the late prenatal trajectories, all tracts except the OR have a large dip in tissue fraction along the tract. In the primary visual cortex, the subplate disappears during the final weeks of gestation, whereas in the somatosensory cortex there are still subplate neurons present in term-born neonates (*Kostovic and Rakic, 1990*) and the subplate of the pre-frontal associative cortex gradually disappears over the 6 postnatal months. Therefore, the peaks of fluid fraction in the frontal and sensory trajectories might reflect the lasting presence of subplate in these areas (*Kostović and Jovanov-Milosević, 2006*; *Kostovic and Rakic, 1990*).

There are several important considerations to our work which may limit the interpretation of our findings. MRI has an inherently low signal:noise ratio as signal attenuation is deliberately introduced and EPI acquisitions are highly susceptible to motion and distortion artefacts. Furthermore, the biophysical properties of the fetal environment introduce a series of additional inter-dependent

challenges for acquiring high-quality dMRI data (*Christiaens et al., 2019a*; *Christiaens et al., 2019b*, *Tournier et al., 2020*). In utero imaging methods must adapt and account for unpredictable, sometimes abrupt motion of fetus, in addition to the motion introduced by maternal breathing. Achieving even coverage of the magnetic coil is challenging due to varying fetal and maternal positions inside the scanner, introducing unique bias fields for every subject. The dHCP acquisition protocol and pre-processing pipelines have incorporated dynamic distortion and motion correction algorithms that were designed to specifically tackle the unique artefacts-associated imaging fetuses (*Deprez et al., 2020*; *Cordero-Grande et al., 2018*; *Cordero-Grande et al., 2019*; *Christiaens et al., 2019a*; *Christiaens et al., 2019b*; *Hutter et al., 2018b*; *Price et al., 2019*). However, even after the extensive correction procedures, there is still some residual motion and distortion in the data. For a more extensive and comprehensive review of challenges associated specifically with in utero diffusion imaging and constructive suggestions to address them, readers are directed to *Christiaens et al., 2019b*.

The conclusions that can be drawn from analysing the diffusion MR signal are limited more generally by the level of noise and lack of directionality in the signal, and therefore we are not able to distinguish between ascending and descending thalamic projections as they develop. The fibre-tracking process itself is also very sensitive to the residual motion and distortion in fetal data, producing highly variable results in terms of streamline count per tract. Therefore, the metrics generally used in this field to investigate fibre bundle morphology, such as streamline count, a proxy for 'innervation density', are not an accurate, quantitative measure (*Calamante, 2019*) and cannot be reliably used to assess fibre morphology. Consequently, we have just used tractography to delineate regions of interest and used the signal contrast in the diffusion maps to quantify microstructure.

The methods described allow the direct study of the maturational effects of the subplate and intermediate zones, which are known to represent critical substrates for early synaptogenesis and the spatial guidance of thalamocortical axons (*Ghosh et al., 1990*). Damage to this essential structural framework for developing cortical circuitry has been implicated in the origins of numerous developmental disorders and is suspected to underly the altered structural and functional connectivity of the thalamus in preterm infants (*Back and Volpe, 1997*; *Volpe, 2001*; *Volpe, 2009*; *Hüppi et al., 2001*; *Counsell et al., 2003*; *Hadders-Algra et al., 2017*; *Mathur and Inder, 2009*; *Toulmin et al., 2015*; *Ball et al., 2012*; *Ball et al., 2015*). It is therefore critical to use clinically relevant tools, such as in utero MRI, to relate the microstructural properties of these transient fetal compartments to neurobiological processes. This improves mechanistic insight about both healthy white matter maturation and the developmental origins of white matter pathologies.

With this study we explore the development of thalamocortical white matter by quantifying microstructure in the different layers of the fetal brain. Using diffusion metrics, we characterise the emergence of structural connectivity from the thalamus to spatially and functionally distinct cortical brain regions. We observe correlations between the transitioning tissue components and key neurobiological processes in white matter development. By providing a detailed normative reference of MR signal change during the second to third trimester, this will help future studies to identify if the tissue properties of specific compartments are affected by preterm birth or other perinatal injury. To this effect, all fetal MRI data is made available to the research community.

## Materials and methods

### Sample

The study was approved by the UK Health Research Authority (Research Ethics Committee reference number: 14/LO/1169) and written parental consent was obtained in every case for imaging and open data release of the anonymised data. All data was acquired in St Thomas Hospital, London, United Kingdom. The sociodemographic characteristics of this sample are representative of the diversity in the London population (see *Appendix 1—figure 1*).

### Acquisition, pre-processing, and quality control

GA was determined by sonography at 12 post-ovulatory weeks as part of routine clinical care. Three-hundred fetal MRI datasets were acquired with a Philips Achieva 3T system, with a 32-channel cardiac coil in maternal supine position. dMRI data was collected with a combined spin echo and field echo (SAFE) sequence (*Hutter et al., 2018a*, *Cordero-Grande et al., 2018*) at 2 mm isotropic resolution,

using a multi-shell diffusion encoding that consists of 15 volumes at b=0 s/mm$^2$, 46 volumes at b=400 s/mm$^2$, and 80 volumes at b=1000 s/mm$^2$ lasting 14 min (*Christiaens et al., 2019a*). For more insight on the choice of b-shell values for this acquisition, readers are directed to *Appendix 1—figure 4* (*Tournier et al., 2019*). The protocol also included the collection of structural T2w, T1w, and fMRI data, for a total imaging time of approximately 45 min (*Price et al., 2019*).

dMRI data were processed using a bespoke pipeline (*Christiaens et al., 2019a*) that includes generalised singular value shrinkage image denoising and debiasing from complex data (*Cordero-Grande et al., 2019*), dynamic distortion correction of susceptibility-induced B0 field changes using the SAFE information (*Ghiglia and Romero, 1994*; *Cordero-Grande et al., 2018*; *Hutter et al., 2018b*) and slice-to-volume motion correction based on a multi-shell spherical harmonics and radial decomposition (SHARD) representation (*Christiaens et al., 2021*). Quality control (QC) was implemented using summary metrics based on the gradient of the motion parameters over time and the percentage of slice dropouts in the data (*Christiaens et al., 2021*). This was followed up with expert visual assessment, which considered any residual or uncorrected artefacts. Image sharpness, residual distortion, and motion artefacts were visually assessed and scored between 0 and 3, with 0 (=failure, e.g. because the subject moved out of the field of view) to 3 (=high quality), based on the mean b=0, b=400, and b=1000 images and the ODFs estimated with MSMT-CSD. See *Appendix 1—figure 2* for examples of subjects that were excluded. Both DWI and T2 for each fetus were required to facilitate co-registration to template space via a structural intermediate. After co-registration, the QC scores were checked and validated again by different authors (MP, AU, SW), only subjects scoring 2 or 3 that were well aligned in T2 space were admitted to this study. Based on the above criteria, 140 of the 300 subjects that were pre-processed were classified as high-quality reconstructions for both DWI and T2 modalities. Post hoc analysis was also conducted to check if any motion-related parameters correlate with GA (*Appendix 1—figure 3*).

## Diffusion modelling and template generation

All diffusion processing and tractography was done using MRtrix3 (*Tournier et al., 2019*). To deconvolve the tissue and fluid components of the diffusion data, white matter and cortical spinal fluid (CSF) response functions were estimated for each subject using T2-based tissue segmentations as inclusion areas. White matter response functions were extracted from areas of relatively mature white matter (corticospinal tract and corpus callosum) using the 'tournier' algorithm and CSF responses using the 'dhollander' algorithm in MRtrix3 (*Jeurissen et al., 2014*; *Tournier et al., 2019*; *Tournier et al., 2013*). The white matter response functions of the oldest 20 subjects were averaged to obtain a group-average response function of relatively mature white matter, whilst a group-average CSF response function was calculated from the whole cohort of subjects. dMRI signal of all subjects was subsequently deconvolved into tissue ODF and fluid components using MSMT-CSD and the group-average white matter and CSF response functions (*Jeurissen et al., 2014*), and resulting components were intensity normalised for each subject (*Raffelt et al., 2011*). Subject ODFs warped into weekly templates through a series of coarse pose normalisation and non-linear diffeomorphic image registration steps (*Jenkinson et al., 2002*; *Raffelt et al., 2011*; *Pietsch, 2018*). These transformations were composed to obtain pairs of inverse consistent diffeomorphic subject-to-template and template-to-subject warps.

## Connectome generation and tractography

The ODF templates were co-registered to the Boston T2-fetal atlas (*Gholipour et al., 2017*) using non-linear registration (*Avants et al., 2008*). The tissue segmentations of the cortex, white matter, and deep grey matter were used for anatomically constrained tractography to generate whole-brain structural connectomes of 100 M streamlines in each gestational week (*Smith et al., 2012*; *Tournier et al., 2019*). The connectomes were filtered down to 10 M streamlines using the SIFT algorithm (*Smith et al., 2013*; *Tournier et al., 2019*), so that the number of streamlines connecting the two regions are approximately proportional to the cross-sectional area of the fibres connecting them (*Smith et al., 2013*). In each weekly template, thalamocortical pathways of interest were defined in both hemispheres by filtering the connectome using seed regions derived from the Boston T2-fetal atlas (*Gholipour et al., 2017*), including the thalamus, primary motor cortex, primary sensory cortex, posterior parietal cortex, dorso-lateral prefrontal cortex, and primary visual cortex. We also used

additional ROIs to exclude spurious streamlines that were projecting away from the expected path of the tract (e.g. to exclude callosal fibres from the TM).

### Tract density parcellation of thalamus

Tckmap was used to identify which voxels in the thalamus mask were traversed by the streamlines of each tract (*Calamante et al., 2010*). The tract density maps were merged using FSL (*Jenkinson et al., 2012*) and a colour-coded parcellation volume was constructed reflecting the maximum density tract for each voxel. For visualisation, the tract density maps for each tract were thresholded at 80%, only to include voxels with the highest streamline connectivity.

### Extracting tissue and fluid fraction values

To extract diffusion metrics for analysis, tracts were transformed from the templates to age-matched subject space to be overlaid onto the normalised fluid ODF and the normalised tissue ODF. The mean value within the segmented tracts was calculated to give the tissue and fluid fractions.

### Microstructural profiling

In each template, thalamocortical tracts were filtered so all the streamlines for each tract were of the same length, to ensure even sampling intervals along them. All template tracts were then registered into a standard space and resampled to 100 points, before being transformed to individual subjects and overlaid on the normalised tissue and fluid fraction maps. The average value for each sampling point was calculated to create a microstructural profile along the path between the thalamus and the cortical plate. To provide a reference for microstructural differences between fetal brain compartments, tracts were overlaid on the atlas-derived tissue parcellations. The value of the tissue labels underlying the tract were used to establish which sampling points corresponded to each fetal compartment. These boundaries between compartments were then used to label the plots in *Figure 5*.

## Acknowledgements

We thank the patients who agreed to participate in this work and the staff of St Thomas' Hospital London. This work was supported by the European Research Council under the European Union Seventh Framework Programme (FP/2007–2013)/ERC Grant Agreement No. 319456. We acknowledge infrastructure support from the National Institute for Health Research (NIHR) Mental Health Biomedical Research Centre (BRC) at South London and Maudsley NHS Foundation Trust, King's College London, and the NIHR-BRC at Guy's and St Thomas' NHS Foundation Trust. We also acknowledge grant support in part from the Wellcome Engineering and Physical Sciences Research Council (EPSRC) Centre for Medical Engineering at King's College London (WT 203148/Z/16/Z) and the Medical Research Council (UK) (MR/K006355/1) and (MR/L011530/1). SW was supported by PhD funding from the UK Medical Research Council/Sackler Foundation (MR/P502108/1). JO is supported by a Sir Henry Dale Fellowship jointly funded by the Wellcome Trust and the Royal Society (206675/Z/17/Z). JO, MR, ADE, JO, and TA received support from the Medical Research Council Centre for Neurodevelopmental Disorders, King's College London (MR/N026063/1). TA was supported by an MRC Clinician Scientist Fellowship (MR/P008712/1). VK and TA were supported by an MRC Transition Support Award (MR/V036874/1). Support for this work was also provided by the NIHR-BRC at Kings College London, Guy's and St Thomas' NHS Foundation Trust in partnership with King's College London, and King's College Hospital NHS Foundation Trust.

# Additional information

## Funding

| Funder | Grant reference number | Author |
|---|---|---|
| European Research Council | Seventh Framework Programme: FP/2007/2013 | Maximilian Pietsch Lucilio Cordero-Grande Daan Christiaens Vyacheslav R Karolis Vanessa Kyriakopoulou Anthony N Price Jana Hutter Emer J Hughes Jacques-Donald Tournier Joseph V Hajnal A David Edwards |
| Wellcome Trust | Sir Henry Dale Fellowship: 206675/Z/17/Z | Jonathan O'Muircheartaigh |
| Medical Research Council Centre for Neurodevelopmental Disorders | MR/N0266063/1 | Siân Wilson Jonathan O'Muircheartaigh Mary A Rutherford A David Edwards Tomoki Arichi |
| Medical Research Council | Translation support fellowship: MR/V036874/1 | Vyacheslav R Karolis Tomoki Arichi |
| Wellcome / EPSRC Centre for Biomedical Engineering, Kings College London | WT 203148/Z/16/Z | Anthony N Price Jana Hutter Jacques-Donald Tournier Joseph V Hajnal |
| Medical Research Council | Clinician Scientist Fellowship MR/P008712/1 | Tomoki Arichi |
| Medical Research Council | MR/K006355/1 | Siân Wilson Maximilian Pietsch Lucilio Cordero-Grande Daan Christiaens Alena Uus Vyacheslav R Karolis Vanessa Kyriakopoulou Kathleen Colford Anthony N Price Jana Hutter Mary A Rutherford Emer J Hughes Serena J Counsell Jacques-Donald Tournier Joseph V Hajnal A David Edwards Jonathan O'Muircheartaigh Tomoki Arichi |
| Medical Research Council | MR/L011530/1 | Serena J Counsell |

The funders had no role in study design, data collection and interpretation, or the decision to submit the work for publication. For the purpose of Open Access, the authors have applied a CC BY public copyright license to any Author Accepted Manuscript version arising from this submission.

## Author contributions

Siân Wilson, Conceptualization, Formal analysis, Validation, Investigation, Visualization, Methodology, Writing – original draft, Writing – review and editing; Maximilian Pietsch, Data curation, Software, Formal analysis, Methodology, Writing – review and editing; Lucilio Cordero-Grande, Data curation, Software, Methodology, Writing – review and editing; Daan Christiaens, Data curation, Formal analysis, Methodology, Writing – review and editing; Alena Uus, Data curation, Methodology; Vyacheslav R Karolis, Formal analysis, Investigation, Methodology, Writing – review and editing; Vanessa

Kyriakopoulou, Kathleen Colford, Anthony N Price, Emer J Hughes, Data curation; Jana Hutter, Data curation, Writing – review and editing; Mary A Rutherford, Data curation, Funding acquisition; Serena J Counsell, Data curation, Investigation, Writing – review and editing; Jacques-Donald Tournier, Data curation, Software, Formal analysis; Joseph V Hajnal, Resources, Data curation, Funding acquisition, Methodology, Project administration; A David Edwards, Supervision, Funding acquisition, Investigation, Methodology, Project administration, Writing – review and editing; Jonathan O'Muircheartaigh, Conceptualization, Resources, Data curation, Supervision, Funding acquisition, Validation, Investigation, Methodology, Project administration, Writing – review and editing; Tomoki Arichi, Conceptualization, Resources, Data curation, Formal analysis, Supervision, Funding acquisition, Validation, Investigation, Methodology, Project administration, Writing – review and editing

### Author ORCIDs
Siân Wilson http://orcid.org/0000-0003-4617-3583
Serena J Counsell http://orcid.org/0000-0002-8033-5673
A David Edwards http://orcid.org/0000-0003-4801-7066
Tomoki Arichi http://orcid.org/0000-0002-3550-1644

### Ethics
Human subjects: The study was approved by the UK Health Research Authority (Research Ethics Committee reference number: 14/LO/1169) and written parental consent was obtained in every case for imaging and open data release of the anonymized data.

### Decision letter and Author response
Decision letter https://doi.org/10.7554/eLife.83727.sa1
Author response https://doi.org/10.7554/eLife.83727.sa2

---

## Additional files

### Supplementary files
• MDAR checklist

### Data availability
Developing Human Connectome project data is open-access and available for download following completion of a data-usage agreement via: http://www.developingconnectome.org/. Data is also available at: https://nda.nih.gov/edit_collection.html?id=3955.

The following dataset was generated:

| Author(s) | Year | Dataset title | Dataset URL | Database and Identifier |
|---|---|---|---|---|
| Uus A, Kyriakopoulou V, Cordero Grande L, Pietsch M, Price A, Wilson S, Patkee P, Karolis V, Schuch A, Gartner A, Williams L, Hughes E, Arichi T, O'Muircheartaigh J, Hutter J, Robinson E, Tournier JD, Rueckert D, Counsell SJ, Rutherford MA, Deprez M, Hajnal JV, Edwards AD | 2023 | Multi-channel spatio-temporal MRI atlas of normal fetal brain development (dHCP project) | https://doi.org/10.12751/g-node.ysgsy1 | G-Node Open Data, 10.12751/g-node.ysgsy1 |

The following previously published dataset was used:

| Author(s) | Year | Dataset title | Dataset URL | Database and Identifier |
|---|---|---|---|---|
| Gholipour A, Rollins CK, Velasco-Annis C, Ouaalam A, Akhondi-Asl A, Afacan O, Ortinau C, Clancy S, Limperopoulos C, Yang E, Estroff JA, Warfield SK | 2017 | A normative spatiotemporal MRI atlas of the fetal brain for automatic segmentation and analysis of early brain growth | http://crl.med.harvard.edu/research/fetal_brain_atlas/ | Harcard CRL/fetal brain atlas, fetal_brain_atlas/ |

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

## Appendix 1

### Extended details about the methodological decision-making process

#### Sociodemographic information

One of the aims of the dHCP was to ensure that the cohort analysed were representative of the diverse sociodemographic spread within the London population. We compared our cohort with information derived from the Mayor of London database (https://data.london.gov.uk/dataset/indices-of-deprivation) to check if the distributions were similar (*Appendix 1—figure 1*). We find that socioeconomic status, according to the index of multiple deprivation, and the ethnicity is comparable between our cohort and London. This suggests that our sample is representative, and our results should be generalisable across the London population. To ensure that we were not observing the effect of a preterm pregnancy, we also provide information on the GA at birth of the fetal cohort and the baby birth weight.

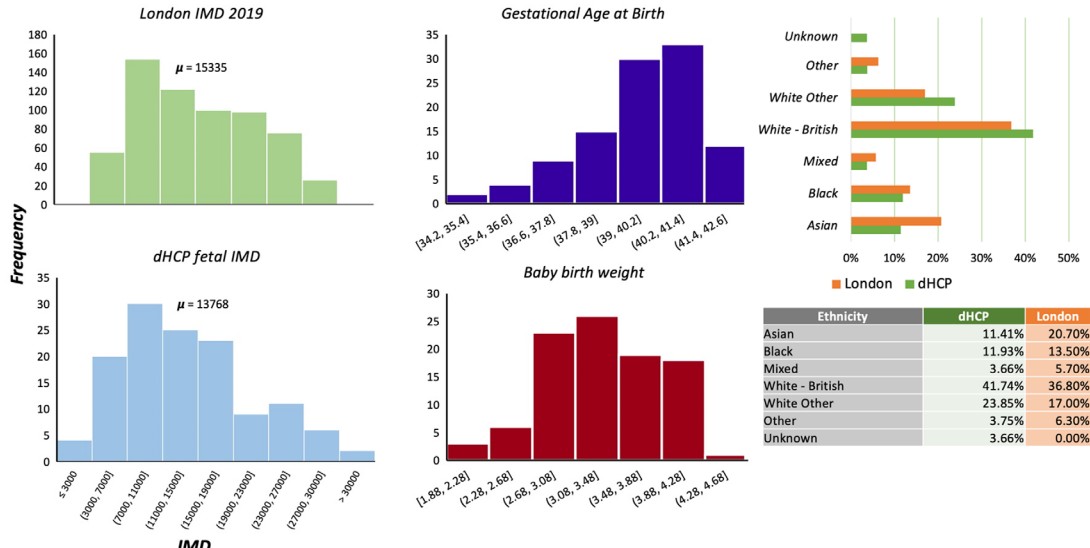

**Appendix 1—figure 1.** Sociodemographic and neonatal follow-up information. Including index of multiple deprivation (IMD), London. Gestational age at birth, birth weight, and ethnicity.

#### Quality control

QC was implemented using summary metrics based on the gradient of the motion parameters over time and the percentage of slice dropouts in the data (*Christiaens et al., 2021*). This was followed up with expert visual assessment, which considered any residual or uncorrected artefacts. Image sharpness, residual distortion, and motion artefacts were visually assessed and scored between 0 and 3, based on the mean b=0, b=400, and b=1000 images. The scoring system was as follows: 0=failure, brain outside field of view, 1=banding and large distortion, 2=blurring but no major artefacts, 3=relatively high quality.

**Score = 0, brain outside FOV**

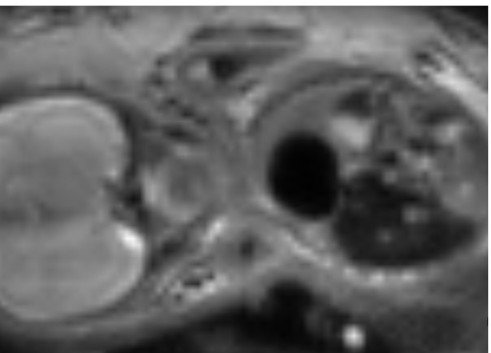

**Score = 1, banding**

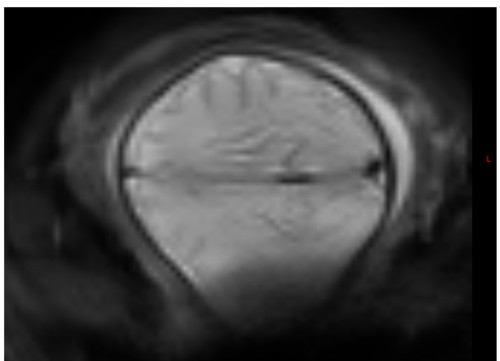

**Score = 2, cortical blurring but no major artefacts**

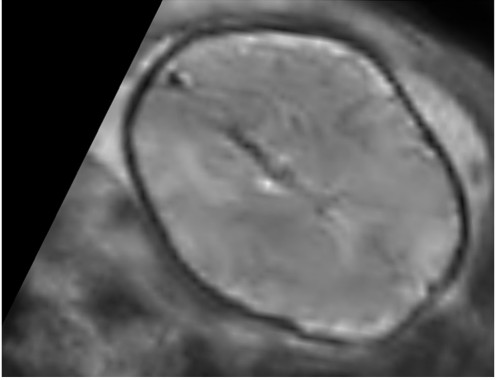

**Score = 3, high quality fetal DWI**

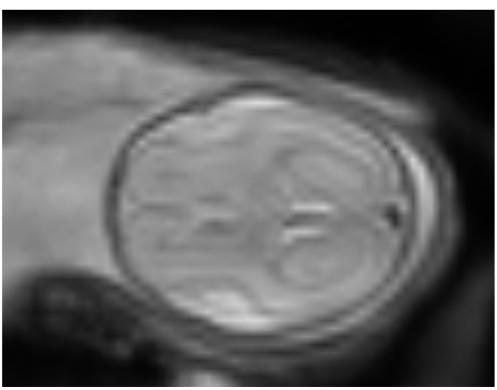

**Appendix 1—figure 2.** Examples of QC scoring scale for subjects, between 0 and 3, with 0 (=failure, e.g. subject moved out of the field of view) to 3 (=high quality).

As an additional check to see if the extent of motion affected our results, we correlated the indices derived from the SHARD pipeline with GA. There are three output parameters per subject, which are signal-to-noise ratio (SNR), rotation, and translation. SNR = mean b=0 signal/mean noise level, measured in the reconstruction mask using MP-PCA denoising. Rotation = mean change in rotation of the subject pose between slices (degrees). Translation = mean change in translation of the subject pose between slices (mm).

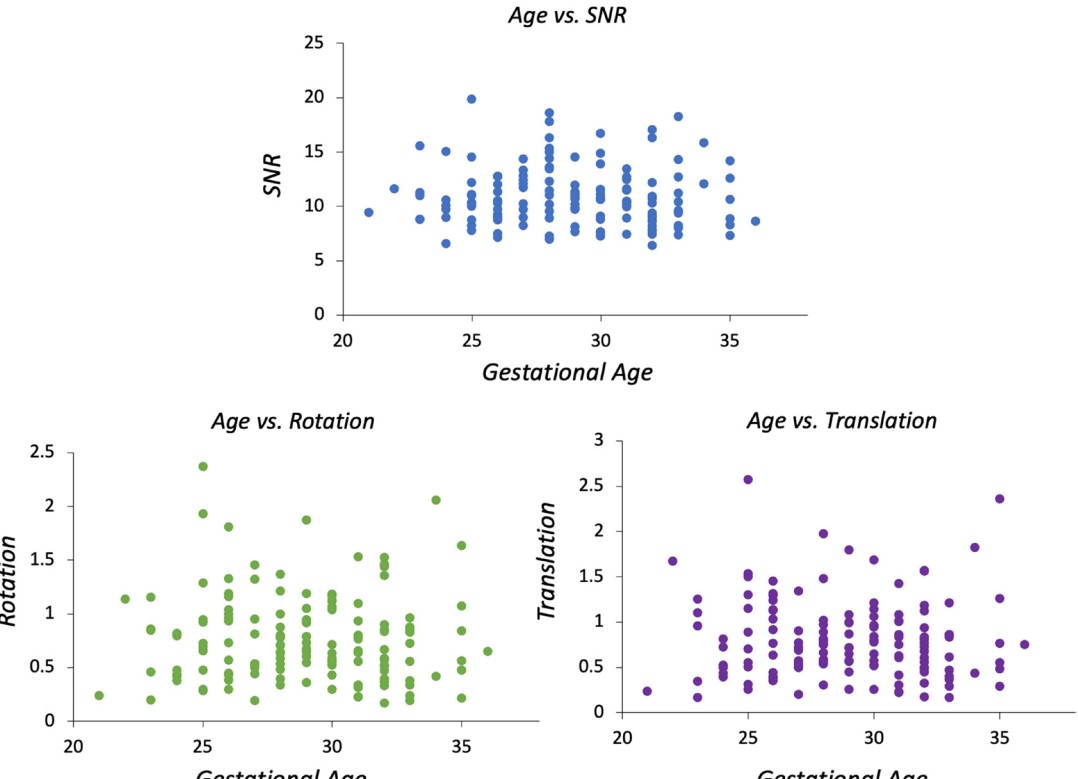

**Appendix 1—figure 3.** Correlations between spherical harmonics and radial decomposition (SHARD)-derived indices describing subject motion and gestational age (signal-to-noise ratio [SNR], rotation and translation).

## Diffusion data modelling

Previous publications provide detail on the optimisation process used to define the b-shell values used (*Tournier et al., 2019* and *Christiaens et al., 2019b*; *Christiaens et al., 2019a*). This work demonstrates that there is a marked loss of signal at higher b-shells (>1000 s/mm³) in the neonatal brain compared to adults, due to the tissue being inherently much less restricted. There is also increased noise at higher b-shells. Both issues are amplified in the fetal brain, there are long relaxation times in the water-rich, relatively unrestricted fetal brain tissues and the amount of noise is higher.

To demonstrate the suitability of MSMT-CSD for this dataset, an important consideration is that the MSMT-CSD approach doesn't mandate a specific set of b-values which are required for the modelling process, only that there are multiple shells in addition to b=0. The technique depends on the signals related to different tissue types having sufficiently distinct behaviours across the b-values sampled, and so depends inherently on the characteristics of the tissue. To demonstrate the distinct properties of the different compartments in the fetal brain, we have now included a figure in the Appendix (*Appendix 1—figure 4*) showing the signal amplitude per tissue type as a function of b-value, and the ODFs after MSMT-CSD has been applied.

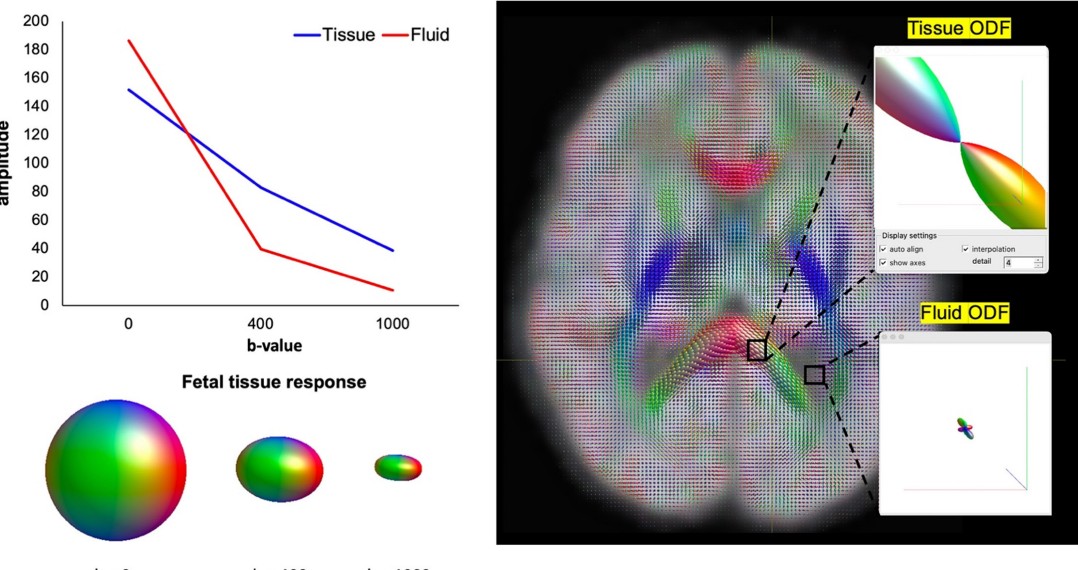

**Appendix 1—figure 4.** Signal amplitude decay in tissue vs. fluid response functions at different b-shells (left). Polar plots of the distinct orientation density functions (ODFs) between the corpus callosum (white matter, tissue compartment) and ventricle (cortical spinal fluid [CSF], fluid compartment) after multi-shell multi-tissue constrained spherical deconvolution (MSMT-CSD) is applied to the data (right).

## Along-tract changes for specific gestational weeks

Prior to grouping the data as seen in *Figure 5*. The analysis was conducted for individual subjects, then averaged for each gestational week. We present these trajectories (*Appendix 1—figures 5 and 6*) to demonstrate how there are gradual changes in the diffusion signal, on a weekly basis, that justify our decision (for easier interpretation) to group the data into early, mid, and late prenatal groups.

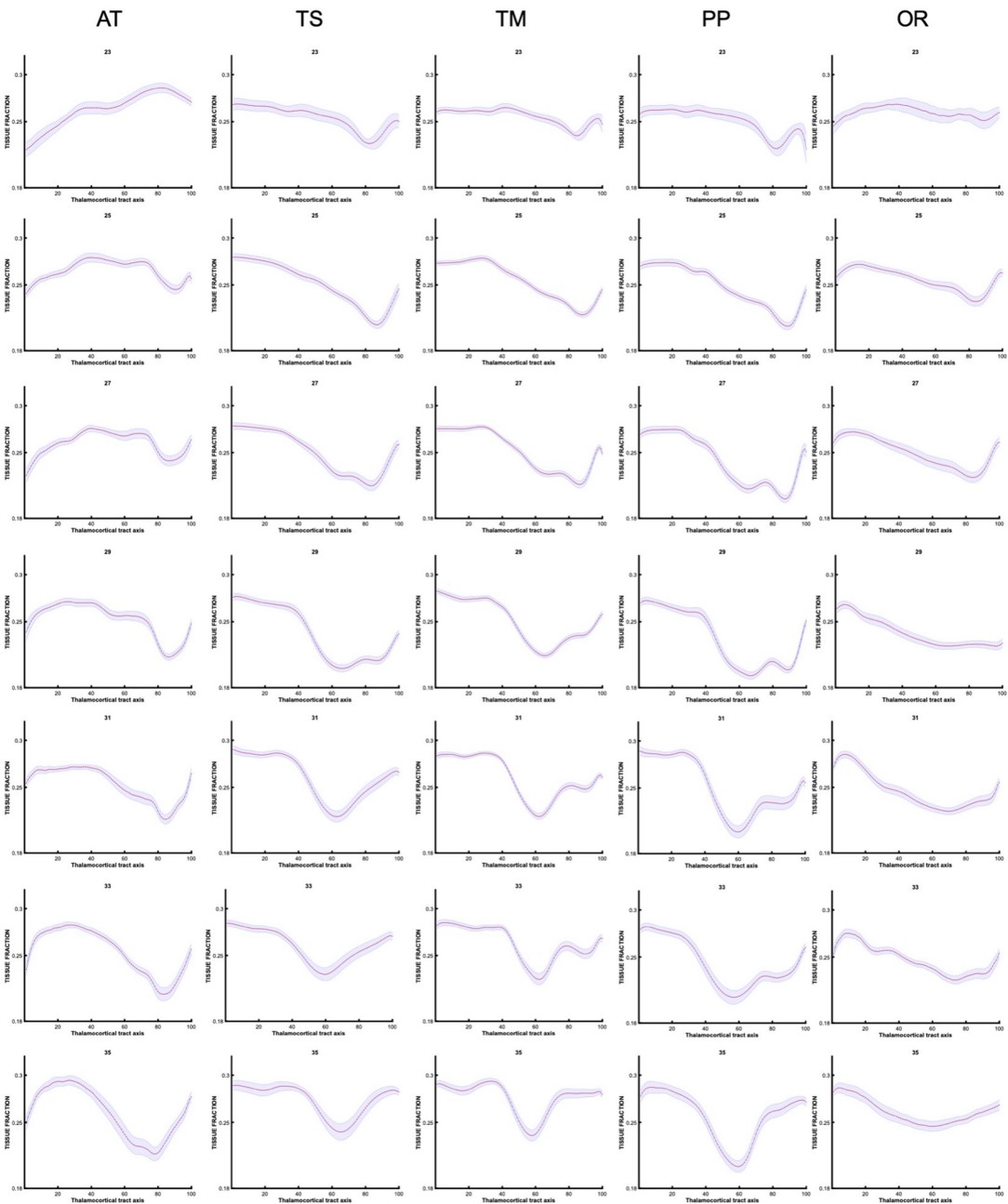

**Appendix 1—figure 5.** Tissue fraction trajectories along thalamocortical tract axis in each gestational week, prior to grouping into early, mid, and late prenatal (every other week shown).

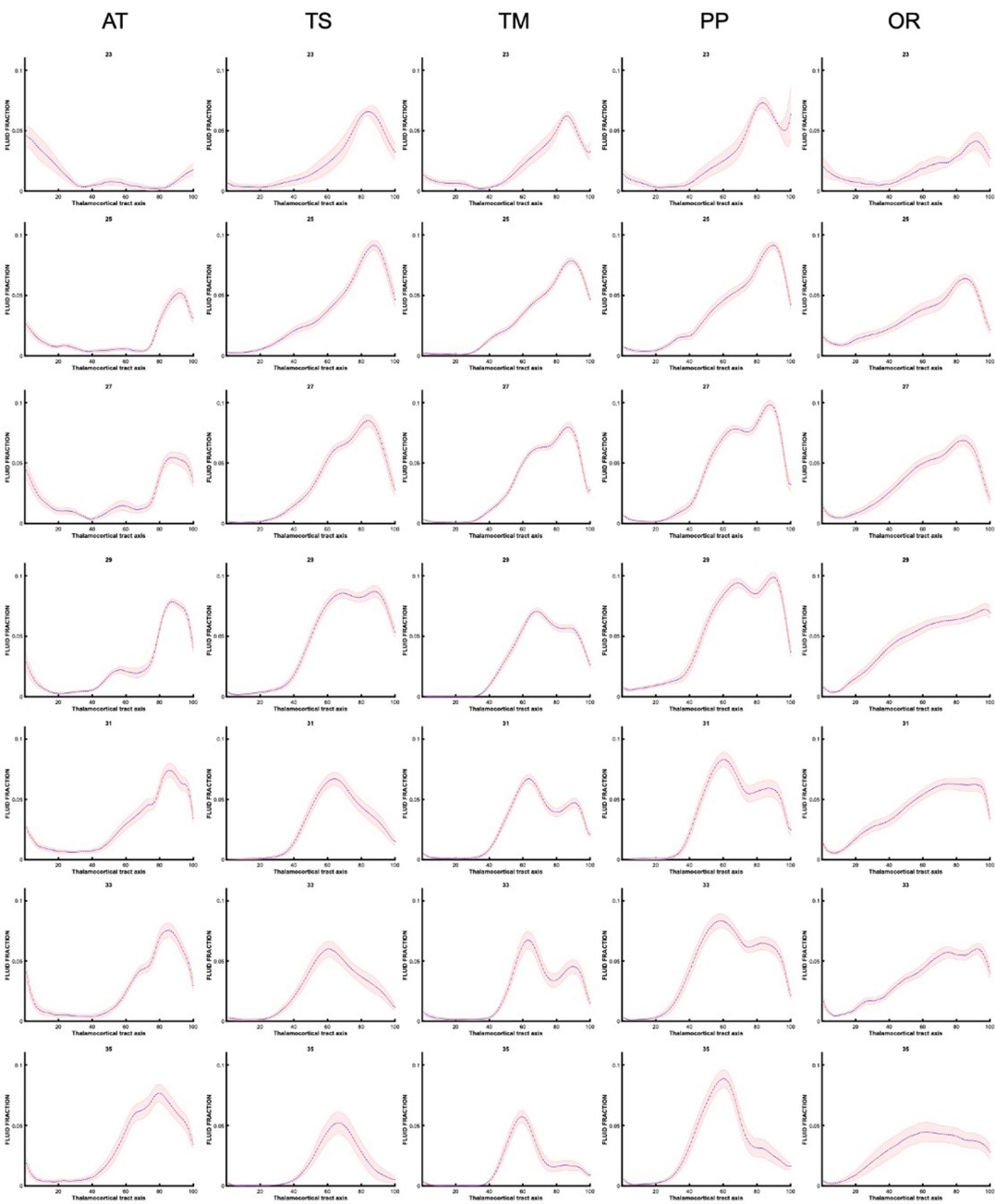

**Appendix 1—figure 6.** Fluid fraction trajectories along thalamocortical tract axis in each gestational week, prior to grouping into early, mid, and late prenatal (every other week shown).

