## [Editor Report]

This study presents important new findings regarding prenatal thalamocortical development. The authors present convincing evidence, while overcoming substantial methodological challenges, in charting prenatal brain development in vivo. This work will be of interest to pediatric and developmental neuroscientists and neuroradiologists.

---

## [Decision Letter]

**Decision letter after peer review:**

Thank you for submitting your article "Spatiotemporal tissue maturation of thalamocortical pathways in the human fetal brain" for consideration by *eLife*. Your article has been reviewed by 3 peer reviewers, including involved in the review of your submission have agreed to reveal their identity: Finnegan as the Reviewing Editor and Reviewer #1, and the evaluation has been overseen by Jonathan Roiser as the Senior Editor. The following individuals involved in the review of your submission have agreed to reveal their identity: Cassandra Hendrix (Reviewer #2); Zoltan Molnar (Reviewer #3).

Essential revisions:

The reviewers were in agreement that the paper represents an impressive accomplishment that will be impactful to the field, and which generates multiple novel insights. Several points were raised for aspects of the manuscript that could be strengthened.

1) Justify and clarify aspects of the methodological approach, including the population characteristics, several aspects of the statistical approaches employed, and potential limitations of the acquisition being applied to this population.

2) Demonstrate and discuss the robustness of the sensitivity of the results presented.

3) Consider further implications of these findings in the context of known anatomy of the thalamus and its development in prior literature.

For further details please see the "Public reviews" and "Recommendations for the authors" below.

*Reviewer #1 (Recommendations for the authors):*

This is an impressive study that overcomes a number of obstacles in performing fetal imaging analysis.

Regarding the application of MSMT-CSD modeling, the b-values used in the current study (400, 1000) seem comparatively low to previous studies employing MSMT modeling which I am aware of. Some clarification regarding the validity of these approaches given the sequences used (e.g., in consideration of Tournier, et al., 2013), e.g., including citing prior studies demonstrating validity for a similar range of b-values, would lend support to the application of these analyses to the current data set.

Assessment of the location-specific developmental effects is accomplished by normalizing each tract on a 0-100 scale between the thalamus and cortex. Does this normalization, in the context of an expanding brain, imply that a given point may be comparatively closer to the terminal zone at different ages? Could this account for (all or some of the) age-related changes in microstructure identified (i.e., if certain features are age-invariant with respect to absolute, rather than relative, position along the tract)? Demonstrating that group differences persist when selecting (e.g.) constant length segments at the ends and middle of each tract would help rule out brain size-induced topographic change as driving these effects (though other statistical approaches could likely also accomplish this).

I'm unsure what statistics support conclusions regarding position-specific differences in tissue fraction. E.g., the discussion summarizes that "the early period is characterized by higher tissue fractions in the middle of the [thalamocortical] tract". While this appears likely based on visual inspection of the tracts in Figure 5, it is not clear what specific statistical analyses support this inference. Further, given that differences were assessed both as a function of GA (though perhaps exclusively by group) and as a function of distance, some degree of multiple comparison correction is probably warranted, but not currently employed (or at least, not described).

*Reviewer #2 (Recommendations for the authors):*

First, I would like to commend the authors on a thoughtful, creative combination of histological maps and dMRI data. One of the major challenges of fetal neuroimaging work is understanding how co-occurring and dynamic neurobiological processes contribute to metrics derived from in vivo imaging. The present study presents an exciting model for contextualizing dMRI findings in relation to what we know about brain development from histology studies. The following recommendations are offered with the hope that they can strengthen an already exceptional paper:

1. Please provide a table with sociodemographic information about the sample. This information is necessary for determining the representativeness of the sample, inferring the generalizability of the present findings, and working towards appropriate diversification of developmental neuroscience as a field. Given that age-related change is the primary question of interest, it would also strengthen the study to show that fetuses of different ages do not systematically differ from one another in terms of other factors that may alter developmental trajectories.

2. The manuscript would benefit from more detail regarding motion correction. The authors describe using slice-to-volume motion correction (SHARD) and implementing quality control based on slice dropouts and gradient of motion parameters. However, additional information is required to enhance transparency and aid future replication efforts. What cut-offs were used to determine whether to include vs. exclude a subject? How many fetuses were excluded based on the selected quality control thresholds? Can the authors please provide more information about the visual assessment that was conducted? Were post-processing motion parameters correlated with tissue or fluid fraction values or with fetal age? It would also be helpful to include visual examples of residual or uncorrected artefacts that resulted in subject exclusion, even if only in the supplemental materials. Finally, Christiaens et al., 2021 is cited several times in the text (including in reference to preprocessing techniques used for correcting head motion) but is not included in the reference list.

3. What size were the along-tract analysis segments and how was this segment size selected? Showing the results replicate (or don't) with different segment sizes would show the robustness of results and could inform an ideal resolution at which to examine tissue maturation in the fetal brain.

4. Additional analyses that show the primary findings do not replicate with noise data but do replicate across different preprocessing choices would enhance rigor, especially since the field of fetal neuroimaging lacks a widely agreed-upon set of best practices. For instance, if segments from the along-tract analysis are scrambled, does the outward-to-inward maturational trajectory disappear? The disappearance of associations with scrambled data would enhance confidence that this finding is not simply a byproduct of the way the data were preprocessed. In addition, showing the results replicate following different preprocessing choices would aid in showing effect robustness.

5. Several studies have identified sex differences in fetal brain maturation. Given the even split in the present sample, examining whether tissue maturation trajectories differ by fetal sex would be informative. I wonder if the authors would consider adding this analysis.

6. The limitations of fetal dMRI should be clearly described in the discussion to aid reader interpretation.

7. It would improve manuscript readability to clarify why the authors believe it is important to report both tissue and fluid fractions. They appear to be reciprocals of one another so the added value of reporting both is unclear.

8. I would avoid subjective phrases like "thalamocortical connections represent the most important inputs into the developing cortex" and instead use more neutral language like "thalamocortical connections play a key role in guiding…"

*Reviewer #3 (Recommendations for the authors):*

Overall, this is a very valuable study with excellent datasets. The material provides a unique and detailed description of the changes in microstructure in the thalamus and cortex. I am very supportive of publication, but I feel that more could have been done to bridge the gap between MRI and histology and to link the results to the basic principles of thalamocortical development.

Criticisms:

1. Using tract density imaging, the authors observed that the cortical areas were connected to specific thalamic regions, organised in an anterior-posterior axis. This anterior-posterior representation of cortical connectivity in the thalamus was consistent with previous descriptions in rodent thalamocortical connectivity in rodent. However, the relationship between the cortical surface and the volume of the developing thalamus is not explained clearly. In their studies, Molnar and Blakemore emphasized that at the time of arrival, the thalamic projections are already arranged according to rough topography. Anterior to posterior movement along the convexity of the cortex represented a medial-to-lateral movement of labelling in the embryonic rodent thalamus. Ventral to dorsal movement in the cortex resulted in an anterior-to-posterior movement within the thalamus https://doi.org/10.1523/JNEUROSCI.18-15-05723.1998; https://doi.org/10.1111/j.1460-9568.2012.08119.x. It would be good to improve Figure 3 and represent the overall pattern. The very general pattern should be described. Could the thalamic volumes be color coded and the same fainter shade used on the convexity of the cortex of the 3D outline of the brain?

2. The study does not attempt to link the thalamic fibre topography to the histological structure of the thalamus. Thalamic nuclei undergo considerable changes at these early stages, e.g. pulvinar increases enormously and the dorsal lateral geniculate nucleus is changing position from dorsolateral to ventromedial. Such changes were already emphasized in the publications of Le Gros Clark and subsequently Rakic. The authors note that the topography of thalamic nuclei and their cortical connectivity is acquired embryonically, but they do not emphasize the considerable changes in this topography at early stages. The in vivo parcellation of the thalamus is only described as part of the fibre topography, no references are made to the changes in relative positions of the thalamic nuclei, and only the adult similarity is emphasized.

3. The density of the thalamocortical innervation is not equal across the developing or adult cortex. E.g. retrogradely tracing thalamic projections from various parts of the cortex using carbocyanine dyes, it was observed that the same sized carbocyanine dye can label 3-5 times more backlabelled thalamic neurons from the putative somatosensory cortex than from the visual cortex (see references in Z. Molnár, Development of Thalamocortical Connections (Springer, 1998)). The methodologies and the data would have enabled the authors to compare the parameters of the various thalamocortical pathways that they resolved. It would have been good to include some results and comments on these area-specific comparisons.

4. The authors found the spatiotemporal changes in the diffusion signal reflected known developmental processes that take place between the early, mid, and late prenatal period. "The early period is characterized by higher tissue fractions in the middle of the tract, where there is a radial scaffold for migrating neurons. As this scaffold dissipates in the mid-prenatal period, this is accompanied by a reduction in the tissue fraction in the middle of the tract, and an increase in towards the termination of the tracts as the neurons of the cortical plate mature." The fact that these changes can be now resolved with in vivo imaging is excellent, but the data itself is under-analysed.

The authors correctly describe the possible anatomical explanations for the changes of the signals: "In terms of the diffusion signal, this strongly aligned microstructure of the radial glia is represented in our results by a higher tissue fraction in the transient compartments containing the most migratory cells (such as the VZ, IZ). Conversely, we observe the lowest tissue fraction in the early prenatal SP, as this compartment predominantly contains hydrophilic extracellular matrix." These are key processes of cortical development and the authors have the material to draw conclusions on the various gradients on the cortical plate and subplate changes. It would have been good to get a feel for these dynamic changes and include some representations of the area-specificity of these changes. Surely the increase of the cortical plate has areal variation and also the increase and resolution of the subplate are also area specific. Measuring and representing these differences would increase the impact of the study. Moreover, it would also open avenues to study left-right and gender differences.

5. The authors have interesting discussions on the corticofugal and thalamocortical connectivity through the internal capsule. "The microstructural change along the anterior thalamic radiation suggests increasing tissue fraction between the deep grey matter, VZ, and IZ. We hypothesise that the high tissue fraction in the IZ is due to densely packed ascending and descending bundles within the anterior limb of the internal capsule." The exact timing of the outgrowth, intermingling, and arrival of the thalamocortical and corticothalamic projections are not known, but there are some suggestions that these are relatively earlier than in rodents (https://doi.org/10.1093/cercor/bhz056; https://doi.org/10.1016/B978-0-444-53860-4.00003-9).

6. A wide range of references from various species are used. This is excellent, but it can be also somewhat confusing. I recommend stating the species explicitly when non-human studies are referenced.

7. The formatting of the references will need attention: e.g. in line 69: (I. Kostović and Judaš 2015; Ivica Kostović and Judaš 2010); In line 257: Hutter, Christiaens, et al. 2018; In lines 275 and 277: Clascá, Rubio-Garrido, and Jabaudon 2012; In line 299: Miyazaki, Song, and Takahashi 2016; in line 361: (Ivica Kostovic and Rakic 1990b); in line 364: (Ivica Kostović and Judaš 2006; Ivica Kostovic and Rakic 1990a). In line 374: "Toulmin cerebral cortex" is mentioned, but no specific reference is cited.

Overall, I am very enthusiastic about this very significant and high-quality human developmental imaging data. I understand that the authors are making all this data available for future studies in the field. Nevertheless, I feel that they are in the best position to present some fundamental initial descriptions firsthand and present them in a format with suitable illustrations that will make it into textbooks on human development.

---

## [Author Response]

Reviewer #1 (Recommendations for the authors):This is an impressive study that overcomes a number of obstacles in performing fetal imaging analysis.Regarding the application of MSMT-CSD modeling, the b-values used in the current study (400, 1000) seem comparatively low to previous studies employing MSMT modeling which I am aware of. Some clarification regarding the validity of these approaches given the sequences used (e.g., in consideration of Tournier, et al., 2013), e.g., including citing prior studies demonstrating validity for a similar range of b-values, would lend support to the application of these analyses to the current data set.

We agree that providing insight about the choice of acquisition parameters would benefit readers, as these were specifically optimised for the unique biophysical properties of the fetal environment. We have now signposted the publications that provide detail on the optimisation process used to define the b-shell values used (Tournier et al., 2019 and Christiaens et al., 2019) more clearly in the text (line 451). In essence, these papers demonstrate that there is a marked loss of signal at higher b-shells (> 1000 s/mm3) in the neonatal brain compared to adults, due to the tissue being inherently much less restricted. There is also increased noise at higher b-shells. Both issues are amplified in the fetal brain, there are long relaxation times in the water-rich, relatively unrestricted fetal brain tissues and the amount of noise is higher.

In terms of the validity of MSMT-CSD for this dataset, an important consideration is that the MSMT-CSD approach doesn't mandate a specific set of b-values which are required for the modelling process, only that there are multiple shells in addition to b = 0. The technique depends on the signals related to different tissue types having sufficiently distinct behaviours across the b-values sampled, and so depends inherently on the characteristics of the tissue. To demonstrate the distinct properties of the different compartments in the fetal brain, we have now included a figure in the Appendix (Appendix 1-Figure 4) showing the signal amplitude per tissue type as a function of b-value, and the ODFs after MSMT-CSD has been applied. We have also provided further detail on the motivation for this choice in the discussion (line 265-271).

There is a general scarcity of literature in the fetal diffusion imaging field, but to support that the MSMT-CSD model is as valid as traditional approaches (DTI), in our previous work (Wilson et al., 2021), we demonstrated a high linear correlation between diffusion tensor metrics and the MSMT-CSD components when whole-tract averages are calculated.

Assessment of the location-specific developmental effects is accomplished by normalizing each tract on a 0-100 scale between the thalamus and cortex. Does this normalization, in the context of an expanding brain, imply that a given point may be comparatively closer to the terminal zone at different ages? Could this account for (all or some of the) age-related changes in microstructure identified (i.e., if certain features are age-invariant with respect to absolute, rather than relative, position along the tract)? Demonstrating that group differences persist when selecting (e.g.) constant length segments at the ends and middle of each tract would help rule out brain size-induced topographic change as driving these effects (though other statistical approaches could likely also accomplish this).

We agree that it is important to consider the effect of increasing fetal brain size when studying maturational changes across gestational age. We have operated under the assumption that the morphology of the tracts is consistent with age and thus that the tracts consistently project from the same origin to termination point in all fetuses (for example, from the thalamus to the primary motor cortex). Therefore, any change in the length of tract across gestational age will remain proportional to the change in overall brain size (or distance between the start and end of the tract) and thus each sampling point will represent the same relative distance from the origin and termination across different fetal ages.

We also used this normalisation approach together with an atlas, to cross-reference our trajectories of diffusion values. By overlaying the tracts on the atlas, and corroborating each sampling point with an atlas region, we characterised the diffusion properties of the different fetal compartments independent of their change in size and maturity across gestation. We have included more detail on this choice in the Results section to clarify it for readers (line 218-220).

I'm unsure what statistics support conclusions regarding position-specific differences in tissue fraction. E.g., the discussion summarizes that "the early period is characterized by higher tissue fractions in the middle of the [thalamocortical] tract". While this appears likely based on visual inspection of the tracts in Figure 5, it is not clear what specific statistical analyses support this inference. Further, given that differences were assessed both as a function of GA (though perhaps exclusively by group) and as a function of distance, some degree of multiple comparison correction is probably warranted, but not currently employed (or at least, not described).

Thank you for this suggestion, we agree that a more formal statistical comparison supports interpretation and strengthens the conclusion about microstructural differences between compartments. We have therefore correlated the tissue fraction at each sampling point with gestational age in the thalamic motor tract to demonstrate that there are significant associations with age. As there are 100 sampling points, we also used strict Bonferroni correction for multiple comparisons. This additional analysis has been mentioned in the Results section (line 233-237) and is shown in Author response image 1.

**Author response image 1. sa2fig1:** 

Reviewer #2 (Recommendations for the authors):First, I would like to commend the authors on a thoughtful, creative combination of histological maps and dMRI data. One of the major challenges of fetal neuroimaging work is understanding how co-occurring and dynamic neurobiological processes contribute to metrics derived from in vivo imaging. The present study presents an exciting model for contextualizing dMRI findings in relation to what we know about brain development from histology studies. The following recommendations are offered with the hope that they can strengthen an already exceptional paper:1. Please provide a table with sociodemographic information about the sample. This information is necessary for determining the representativeness of the sample, inferring the generalizability of the present findings, and working towards appropriate diversification of developmental neuroscience as a field. Given that age-related change is the primary question of interest, it would also strengthen the study to show that fetuses of different ages do not systematically differ from one another in terms of other factors that may alter developmental trajectories.

We thank the reviewer for their enthusiasm about our work, and for highlighting the absence of this important information. We have included a sentence in the methods section (line 440) about the recruitment process, where this data was acquired and sociodemographic information about the cohort in the Appendix 1-Figure 1 (including histograms to demonstrate that IMD and ethnicity distributions are well matched to the diverse population in London). We have also included Author response image 2 to demonstrate that there is no significant relationship between IMD and gestational age, suggesting that this is not influencing the microstructural trajectories we observe. When the dataset becomes publicly available later this year, a supporting publication will also be released containing all this information.

2. The manuscript would benefit from more detail regarding motion correction. The authors describe using slice-to-volume motion correction (SHARD) and implementing quality control based on slice dropouts and gradient of motion parameters. However, additional information is required to enhance transparency and aid future replication efforts. What cut-offs were used to determine whether to include vs. exclude a subject? How many fetuses were excluded based on the selected quality control thresholds? Can the authors please provide more information about the visual assessment that was conducted? Were post-processing motion parameters correlated with tissue or fluid fraction values or with fetal age? It would also be helpful to include visual examples of residual or uncorrected artefacts that resulted in subject exclusion, even if only in the supplemental materials. Finally, Christiaens et al., 2021 is cited several times in the text (including in reference to preprocessing techniques used for correcting head motion) but is not included in the reference list.

We appreciate that for reproducibility, more detail about precise exclusion criteria would benefit the reader. The methods section now contains additional information about this process (line 464-474), and the appendix includes examples of subjects that were excluded due to motion and residual artefacts (Appendix 1 – Figure 2).

We have also included correlations between motion-related parameters extracted from the SHARD pipeline, (translation, rotation and SNR) (Christiaens et al., 2021) and gestational age, demonstrating no relationship with age. This is included in Appendix 1-Figure 3. Thank you for pointing out the referencing error, we have now corrected it. We would also like to highlight that when this dataset becomes publicly available later this year, QC scores and motion parameters will also be released that will enhance reproducibility for future studies.

3. What size were the along-tract analysis segments and how was this segment size selected? Showing the results replicate (or don't) with different segment sizes would show the robustness of results and could inform an ideal resolution at which to examine tissue maturation in the fetal brain.

The normalisation approach we used was to standardise the sampling interval along all the tracts to 100 points between the origin and termination. This was to ensure consistency across the cohort, account for changing brain size and image resolution. We have assumed that the morphology of the tracts remains consistent across all ages, as we were able to delineate tracts even in the youngest fetuses which projected from the same origin to termination points (for example, from the thalamus to the primary motor cortex). As the start and end points of the tracts are the same regardless of gestational age, any maturational change in length is proportional to a corresponding change in brain size. The smooth transition between each segment, as demonstrated by the consistency of the curves and their gradual shift with age, suggests that this sampling interval is appropriate for the aims of this study. The error bars shown on Figure 5 that reflect the standard error of the mean also support this, as they demonstrate minimal variation between subjects and a lack of overlap between sections in each of the time windows.

4. Additional analyses that show the primary findings do not replicate with noise data but do replicate across different preprocessing choices would enhance rigor, especially since the field of fetal neuroimaging lacks a widely agreed-upon set of best practices. For instance, if segments from the along-tract analysis are scrambled, does the outward-to-inward maturational trajectory disappear? The disappearance of associations with scrambled data would enhance confidence that this finding is not simply a byproduct of the way the data were preprocessed. In addition, showing the results replicate following different preprocessing choices would aid in showing effect robustness.

We agree with the reviewer that an important caveat in the field of fetal imaging is the lack of a standardised set of procedures with which to conduct analysis of MR data. However, we would like to emphasise that the pre-processing pipeline for the dHCP data has been optimised through an extensive and rigorous procedure, such that the methods used for the reconstruction, motion correction and distortion correction produce the highest quality data (more detail on this procedure is available in the following publications cited in the article: Christiaens et al., 2019, 2021, Cordero-Grande 2019, Pietsch et al., 2019). Please see Appendix 1-Figure 3, where we use parameters derived from the SHARD reconstruction pipeline to demonstrate there are no correlations between motion and gestational age. This figure also highlights that for fetuses of the same age, there is a broad distribution of SNR/translation/slice weight values. We feel that this enhances confidence in our results, because the differences between SNR within a gestational week are not reflected in the tissue and fluid fraction values, which in contrast are consistent for subjects of the same age.

In reference to the suggestion to scramble the segments, we feel that shuffling the values presented in figure 5 would lead to disappearance of the trajectory because of the non-overlapping error bars between compartments at each age. To strengthen our analysis, we have taken on board the initial lack of statistical testing and computed correlations between each sampling point tissue fraction and gestational age and corrected for multiple comparisons (see above, response to reviewer one).

5. Several studies have identified sex differences in fetal brain maturation. Given the even split in the present sample, examining whether tissue maturation trajectories differ by fetal sex would be informative. I wonder if the authors would consider adding this analysis.

We agree this would be beneficial to include and have tested for statistical significance between male and female in the average values across the whole tract. We found there was no significant difference between the average diffusion values in any of the tracts. Please see a statement about this finding in the Results section (line 189).

6. The limitations of fetal dMRI should be clearly described in the discussion to aid reader interpretation.

We agree this is a useful suggestion to help to contextualise the study and the methods used. We have now included a specific limitations section in the discussion (line 388-425) and refer reviewers to a previous publication that comprehensively describes these challenges (Christiaens et al., 2019).

7. It would improve manuscript readability to clarify why the authors believe it is important to report both tissue and fluid fractions. They appear to be reciprocals of one another so the added value of reporting both is unclear.

Thank you for highlighting this point. When fitting this model, the two components are reciprocals of one another if the modelling method has fit the data well and the normalization procedure has worked, so we include this to aid reader interpretation of the results and confirm that data validity. To make our rationale clearer to readers, we have included a sentence in the Results section (line 203-205).

8. I would avoid subjective phrases like "thalamocortical connections represent the most important inputs into the developing cortex" and instead use more neutral language like "thalamocortical connections play a key role in guiding…"

We thank the reviewer for this suggestion and agree more neutral language is appropriate. This suggestion has now been included in the revised article (line 50).

Reviewer #3 (Recommendations for the authors):Overall, this is a very valuable study with excellent datasets. The material provides a unique and detailed description of the changes in microstructure in the thalamus and cortex. I am very supportive of publication, but I feel that more could have been done to bridge the gap between MRI and histology and to link the results to the basic principles of thalamocortical development.Criticisms:1. Using tract density imaging, the authors observed that the cortical areas were connected to specific thalamic regions, organised in an anterior-posterior axis. This anterior-posterior representation of cortical connectivity in the thalamus was consistent with previous descriptions in rodent thalamocortical connectivity in rodent. However, the relationship between the cortical surface and the volume of the developing thalamus is not explained clearly. In their studies, Molnar and Blakemore emphasized that at the time of arrival, the thalamic projections are already arranged according to rough topography. Anterior to posterior movement along the convexity of the cortex represented a medial-to-lateral movement of labelling in the embryonic rodent thalamus. Ventral to dorsal movement in the cortex resulted in an anterior-to-posterior movement within the thalamus https://doi.org/10.1523/JNEUROSCI.18-15-05723.1998; https://doi.org/10.1111/j.1460-9568.2012.08119.x. It would be good to improve Figure 3 and represent the overall pattern. The very general pattern should be described. Could the thalamic volumes be color coded and the same fainter shade used on the convexity of the cortex of the 3D outline of the brain?

We are grateful that this missing but extremely important element of our discussion has been highlighted and have now modified the manuscript accordingly, please see discussion (line 286). We have also adjusted Figure 3 to highlight the anterior-posterior connectivity gradient across the cortex and shaded the different nuclei accordingly. In addition, we have changed the colours of the tracts in Figure 2 according to their anterior-posterior connectivity to clearly demonstrate this pattern.

2. The study does not attempt to link the thalamic fibre topography to the histological structure of the thalamus. Thalamic nuclei undergo considerable changes at these early stages, e.g. pulvinar increases enormously and the dorsal lateral geniculate nucleus is changing position from dorsolateral to ventromedial. Such changes were already emphasized in the publications of Le Gros Clark and subsequently Rakic. The authors note that the topography of thalamic nuclei and their cortical connectivity is acquired embryonically, but they do not emphasize the considerable changes in this topography at early stages. The in vivo parcellation of the thalamus is only described as part of the fibre topography, no references are made to the changes in relative positions of the thalamic nuclei, and only the adult similarity is emphasized.

Thank you for pointing out that this key feature of thalamic development was missing from our discussion. With the resolution (8mm^3^) and image contrast of our images, smaller changes in the position or size of individual nuclei are unfortunately difficult to evaluate though we agree that being able to resolve these changes concurrently would further reinforce our findings. We have now added more on the histological structure of the thalamus to our discussion (line 305-310).

3. The density of the thalamocortical innervation is not equal across the developing or adult cortex. E.g. retrogradely tracing thalamic projections from various parts of the cortex using carbocyanine dyes, it was observed that the same sized carbocyanine dye can label 3-5 times more backlabelled thalamic neurons from the putative somatosensory cortex than from the visual cortex (see references in Z. Molnár, Development of Thalamocortical Connections (Springer, 1998)). The methodologies and the data would have enabled the authors to compare the parameters of the various thalamocortical pathways that they resolved. It would have been good to include some results and comments on these area-specific comparisons.

Thank you for this suggestion. While this would be an interesting question to address, unfortunately we feel that tractography itself (particularly in fetal data) cannot be reliably used to assess fibre morphology. The diffusion MR signal has no directionality, and therefore we are unfortunately not able to distinguish between ascending and descending thalamic projections as they develop. We also find that metrics generally used in this field to investigate fibre bundle morphology, such as streamline count, a proxy for ‘innervation density’, are not an accurate, quantitative measure. These are outlined well in this review article: https://www.ncbi.nlm.nih.gov/pmc/articles/PMC6787694/. Consequently, we have avoided using the data in this way, and just used tractography to delineate regions of interest. We have now added a limitations section to the article and mention this constraint of diffusion methodology specifically (line 388).

4. The authors found the spatiotemporal changes in the diffusion signal reflected known developmental processes that take place between the early, mid, and late prenatal period. "The early period is characterized by higher tissue fractions in the middle of the tract, where there is a radial scaffold for migrating neurons. As this scaffold dissipates in the mid-prenatal period, this is accompanied by a reduction in the tissue fraction in the middle of the tract, and an increase in towards the termination of the tracts as the neurons of the cortical plate mature." The fact that these changes can be now resolved with in vivo imaging is excellent, but the data itself is under-analysed.The authors correctly describe the possible anatomical explanations for the changes of the signals: "In terms of the diffusion signal, this strongly aligned microstructure of the radial glia is represented in our results by a higher tissue fraction in the transient compartments containing the most migratory cells (such as the VZ, IZ). Conversely, we observe the lowest tissue fraction in the early prenatal SP, as this compartment predominantly contains hydrophilic extracellular matrix." These are key processes of cortical development and the authors have the material to draw conclusions on the various gradients on the cortical plate and subplate changes. It would have been good to get a feel for these dynamic changes and include some representations of the area-specificity of these changes. Surely the increase of the cortical plate has areal variation and also the increase and resolution of the subplate are also area specific. Measuring and representing these differences would increase the impact of the study. Moreover, it would also open avenues to study left-right and gender differences.

We appreciate that this is a very rich dataset and there are a lot of questions that we have not addressed or analysed within this paper. Incorporating measures of areal variation in cortical plate development will require surface projection to calculate measures such as cortical thickness and sulcal depth reliably across subjects. Unfortunately, the required processing is non-trivial for fetal MRI data and requires careful optimisation, and thus we felt was beyond the scope of this work. Work is thus currently underway to explore regional differences across the brain for a separate publication that will use a different approach with broader coverage across the whole brain, allowing us to examine areal trends more comprehensively. We will also plan to eventually release surface projected data with the dHCP data release. We imagine this subsequent study will support the findings presented in this paper however, we did not want to overcomplicate the results as we aimed to investigate thalamocortical development more specifically.

5. The authors have interesting discussions on the corticofugal and thalamocortical connectivity through the internal capsule. "The microstructural change along the anterior thalamic radiation suggests increasing tissue fraction between the deep grey matter, VZ, and IZ. We hypothesise that the high tissue fraction in the IZ is due to densely packed ascending and descending bundles within the anterior limb of the internal capsule." The exact timing of the outgrowth, intermingling, and arrival of the thalamocortical and corticothalamic projections are not known, but there are some suggestions that these are relatively earlier than in rodents (https://doi.org/10.1093/cercor/bhz056; https://doi.org/10.1016/B978-0-444-53860-4.00003-9).

We thank the reviewer for this suggestion and are grateful for them highlighting this incredibly relevant study for our discussion. Please see discussion (line 373).

6. A wide range of references from various species are used. This is excellent, but it can be also somewhat confusing. I recommend stating the species explicitly when non-human studies are referenced.

We have prefaced references to non-human studies with a description of the specific species in the revised manuscript to make it clearer for readers.

7. The formatting of the references will need attention: e.g. in line 69: (I. Kostović and Judaš 2015; Ivica Kostović and Judaš 2010); In line 257: Hutter, Christiaens, et al. 2018; In lines 275 and 277: Clascá, Rubio-Garrido, and Jabaudon 2012; In line 299: Miyazaki, Song, and Takahashi 2016; in line 361: (Ivica Kostovic and Rakic 1990b); in line 364: (Ivica Kostović and Judaš 2006; Ivica Kostovic and Rakic 1990a). In line 374: "Toulmin cerebral cortex" is mentioned, but no specific reference is cited.

Thank you for pointing out these inconsistences, we have corrected them and corrected the bibliography in the revised manuscript.

Overall, I am very enthusiastic about this very significant and high-quality human developmental imaging data. I understand that the authors are making all this data available for future studies in the field. Nevertheless, I feel that they are in the best position to present some fundamental initial descriptions firsthand and present them in a format with suitable illustrations that will make it into textbooks on human development.